



# Chemical Characterization of Secondary Organic Aerosol at a Rural Site in the Southeastern U.S.: Insights from Simultaneous HR-ToF-AMS and FIGAERO-CIMS Measurements

Yunle Chen[1], Masayuki Takeuchi[2], Theodora Nah[1,3], Lu Xu[4,5], Manjula R. Canagaratna[6], Harald Stark[6,7], Karsten Baumann[8], Francesco Canonaco[9], André S. H. Prévôt[9], L. Gregory Huey[1], Rodney J. Weber[1], Nga Lee Ng[1,2,4,*]

1 School of Earth and Atmospheric Sciences, Georgia Institute of Technology, Atlanta, GA 30332, USA
2 School of Civil and Environmental Engineering, Georgia Institute of Technology, Atlanta, GA 30332, USA
3 Now at School of Energy and Environment, City University of Hong Kong, Hong Kong SAR, China
4 School of Chemical and Biomolecular Engineering, Georgia Institute of Technology, Atlanta, GA 30332, USA
5 Now at Division of Geological and Planetary Sciences, California Institute of Technology, Pasadena, CA 91125, USA
6 Aerodyne Research, Inc., Billerica, MA 02138, USA
7 Department of Chemistry, University of Colorado at Boulder, Boulder, CO 80309, USA
8 Department of Environmental Sciences and Engineering, Gillings School of Global Public Health, The University of North Carolina at Chapel Hill, Chapel Hill, North Carolina 27599, USA
9 Laboratory of Atmospheric Chemistry, Paul Scherrer Institute, Villigen 5232, Switzerland

*Correspondence to*: Nga Lee Ng (ng@chbe.gatech.edu)

**Abstract.** The formation and evolution of secondary organic aerosol (SOA) was investigated at Yorkville, GA, in late summer (mid-August ~ mid-October, 2016). Organic aerosol (OA) composition was measured using two on-line mass spectrometry instruments, the high-resolution time-of-flight aerosol mass spectrometer (AMS) and the Filter Inlet for Gases and AEROsols coupled to a high-resolution time-of-flight iodide-adduct chemical ionization mass spectrometer (FIGAERO-CIMS). Through analysis of speciated organics data from FIGAERO-CIMS and factorization analysis of data obtained from both instruments, we observed notable SOA formation from isoprene and monoterpenes during both day and night. Specifically, in addition to isoprene epoxydiols (IEPOX) uptake, we identified isoprene SOA formation via hydroxyl hydroperoxide oxidation (ISOPOOH oxidation via non-IEPOX pathways) and isoprene organic nitrate formation via photooxidation in the presence of





NO$_x$ and nitrate radical oxidation. Monoterpenes were found to be the most important SOA precursors at night. We observed significant contributions from highly-oxidized acid-like compounds to the aged OA factor from FIGAERO-CIMS. Taken together, our results showed that FIGAERO-CIMS measurements are highly complementary to the extensively used AMS factorization analysis, and

together they provide more comprehensive insights into OA sources and composition.

## 1 Introduction

Organic aerosol (OA), known for its complexity, represents a substantial fraction of tropospheric submicron aerosol (Kanakidou et al., 2005;Zhang et al., 2007;Kroll and Seinfeld, 2008;Jimenez et al., 2009). Global and regional measurements have revealed that the majority of OA can be secondary in

nature (Lim and Turpin, 2002;Zhang et al., 2007;Weber et al., 2007;Lanz et al., 2007;Huang et al., 2014). The southeastern United States (U.S.) is known for its large biogenic volatile organic compound (VOC) emissions from both conifer and deciduous forests, under the influence of intensive anthropogenic activities (Weber et al., 2007;Xu et al., 2015a). Isoprene and monoterpenes (α-pinene, β-pinene, and limonene) are the most dominant biogenic VOC and SOA precursors in the southeastern

U.S. and there is substantial interest in these compounds. For isoprene-derived SOA, isoprene epoxydiols (IEPOX) uptake followed by subsequent condensed-phase reactions (Surratt et al., 2010;Lin et al., 2012;Paulot et al., 2009) is known to be the major pathway in the southeastern U.S., approximately contributing 18 – 36 % to total OA in warm seasons (Budisulistiorini et al., 2013;Hu et al., 2015;Xu et al., 2015a;Xu et al., 2015b). Isoprene organic nitrates formed from both photooxidation

and nitrate radical oxidation have been characterized in ambient measurements and included in models (Lee et al., 2016;Bates and Jacob, 2019), as well as non-IEPOX SOA formed from hydroxy hydroperoxide (ISOPOOH) oxidation (Krechmer et al., 2015;Nagori et al., 2019). Monoterpene nocturnal reactions have been shown to be an important source of particulate organic nitrates in the southeastern U.S. (Xu et al., 2015a;Xu et al., 2015b;Pye et al., 2015), while more recent studies have

demonstrated that monoterpenes are also the prominent source of total OA in the southeastern U.S. given the large fraction of non-nitrogen-containing monoterpene-derived species (Zhang et al., 2018;Xu et al., 2018).



A better understanding of OA composition is aided by advances in state-of-art real-time aerosol instrumentation in the past two decades. Each instrument, with its unique capabilities, provides one piece of information to the SOA puzzle. The high-resolution time-of-flight aerosol mass spectrometer (HR-ToF-AMS, Aerodyne; henceforth referred to as AMS) (DeCarlo et al., 2006;Canagaratna et al., 2007), for example, has been widely used in both laboratory experiments and field measurements. Designed to quantitatively characterize chemical composition of submicron non-refractory (NR-PM$_1$) aerosol, the AMS produces ensemble average mass spectra for organic and inorganic species. Different methods have been used to deconvolve AMS OA mass spectra, e.g., multiple component analysis (Zhang et al., 2007), positive matrix factorization (PMF) (Ulbrich et al., 2009;Canonaco et al., 2013). Oxygenated OA (OOA) is a subgroup, or factor, that has been ubiquitously resolved by AMS factorization analysis and normally used as a surrogate for secondary OA (SOA), while other OA factors can be more regional and seasonal, e.g., isoprene-derived OA (Isoprene-OA) and biomass burning OA (BBOA) (Jimenez et al., 2009;Ng et al., 2010;Hu et al., 2015;Xu et al., 2015a;Cubison et al., 2011). OOA can be further divided into more-oxidized OOA (MO-OOA, characterized by higher O:C ratio) and less-oxidized OOA (LO-OOA, characterized by lower O:C ratio) (Setyan et al., 2012;Xu et al., 2015a), which have also been named as low-volatility OOA (LV-OOA) and semi-volatile OOA (SV-OOA), respectively, in some studies (Ng et al., 2010;Jimenez et al., 2009). In general, LO-OOA corresponds to fresh SOA and MO-OOA corresponds to aged SOA (Zhang et al., 2005;Zhang et al., 2007;Jimenez et al., 2009;Ng et al., 2010). The two OOA factors account for a large fraction of submicron OA worldwide (Jimenez et al., 2009), but the sources of LO-OOA and MO-OOA at different locations are still largely unknown. The Chemical Ionization Mass Spectrometer (henceforth referred to as CIMS) is a well-established technique for online measurements of gaseous species (Huey, 2007), and the recent combination of a Filter Inlet for Gases and AEROsols (henceforth referred to as FIGAERO) to the CIMS (henceforth referred to as FIGAERO-CIMS) allows for the application of CIMS in aerosol molecular composition characterization (Lopez-Hilfiker et al., 2014). Source apportionment analysis has been performed on CIMS gas- and particle-phase measurements in previous studies in a similar manner to that of AMS measurements (Yan et al., 2016;Massoli et al., 2018;Lee et





al., 2018). Compared to traditional AMS source apportionment, FIGAERO-CIMS can provide more information on the identity of each factor, e.g., chemical formulae of tracer molecules and the location of the maximum desorption signal in temperature space ($T_{max}$), by which enthalpy of sublimation and compound vapor pressure can be evaluated (Lopez-Hilfiker et al., 2014). The FIGAERO-CIMS is

highly complementary to the AMS and could substantially expand our knowledge of the AMS OA factors that have been known for over a decade.

Here, we present results from two-month measurements at Yorkville, GA, a rural site in the southeastern U.S., during a transitional season from summer to fall. Along with a suite of additional instrumentation

(see Nah et al. (2018a;2018b)), AMS and FIGAERO-CIMS were deployed, and factorization analysis was applied to measurements from both instruments, in an effort to gain new insights into established AMS OA factors. By combining AMS and FIGAERO-CIMS measurements, we show that isoprene and monoterpenes were dominant OA precursors during both day and night. We also identify notable isoprene oxidation pathways, besides IEPOX uptake, and their contribution to particulate organic

nitrates, which was less recognized by previous AMS measurements.

## 2 Method

### 2.1 Site description

The ambient measurements took place from mid-August to mid-October 2016 at the South Eastern Aerosol Research and Characterization (SEARCH) field site at Yorkville, Georgia (33.92833 N,

85.04555 W, 394 masl). The instruments were housed in an air-conditioned trailer. The Yorkville site was a long-term field site located in a rural environment approximately 55 km northwest of Atlanta, immediately surrounded by forests and open pastures for cattle grazing. Compared to previous measurements at this site (Xu et al., 2015a;2015b), the sampling period of this study was characterized by a transition from warmer to colder season, which had a direct influence on biogenic VOC emissions.

More details of this 2016 Yorkville campaign have been presented in recent publications by Nah et al. (2018a;2018b).


## 2.2 Instrumentation

An AMS (DeCarlo et al., 2006;Canagaratna et al., 2007) was used to characterize the composition of NR-PM$_1$. Ambient air was sampled through a URG PM$_1$ cyclone at 16.7 L min$^{-1}$ to remove coarse particles. A nafion dryer was placed upstream of the AMS to dry the particles (RH < 20 %) in order to

eliminate the influence of RH on particle collection efficiency (CE) in the AMS (Matthew et al., 2008;Middlebrook et al., 2012). Measurements were taken every minute and post-averaged to a 5-minute time interval. Gas-phase interference was eliminated by subtracting the signals when the AMS sampled through a HEPA filter. Ionization efficiency (IE) calibrations were performed with 300 nm ammonium nitrate particles, and sulfate relative ionization efficiency (RIE) calibrations were performed

with 300 nm ammonium sulfate particles. Both calibrations were conducted on a weekly basis. AMS data were analyzed using the data analysis toolkit SQUIRREL (v1.57) and PIKA (v1.16G) within the Igor Pro software (v6.37, Wavemetrics, Portland, OR). The organics data matrix and error matrix for source apportionment analysis were also generated from PIKA v1.16G. Elemental ratios, including oxygen-to-carbon ratio (O:C), hydrogen-to-carbon ratio (H:C), and nitrogen-to-carbon ratio (N:C),

were obtained using the method outlined by Canagaratna et al. (2015). By comparing AMS with parallel particle-into-liquid sampler (PILS) coupled to ion chromatograph (IC) and filter measurements, a constant CE of 0.9 was applied to AMS measurements (Nah et al., 2018a).

An iodide-adduct FIGAERO-CIMS was used to characterize particle-phase multifunctional organic

species, given the advantage of its high selectivity towards highly-polarizable species, such as carboxylic acids and polyols. A detailed description of FIGAERO-CIMS can be found in Lopez-Hilfiker et al. (2014), while a detailed description of the iodide ionization mechanisms can be found in Huey et al. (1995) and Lee et al. (2014). In brief, ambient air was sampled through a URG PM$_1$ cyclone and PM$_1$ particles were collected on a perfluorotetrafluoroethylene (PTFE) filter (2 µm pore size

Zefluor™, Pall Corporation) in the FIGAERO unit for 25 minutes at a flow rate of 16.7 L min$^{-1}$. To prevent potential positive artefact arising from gases sticking onto the filter during sampling, a 30-cm long parallel plate activated carbon denuder (Eatough et al., 1993) was installed upstream of the FIGAERO inlet. After collection, particles were immediately desorbed off the PTFE filter by heated N$_2$



flowing through the filter. The thermal desorption process took 35 minutes, during which the temperature was increased from room temperature (~ 25 °C) to ~200 °C in 15 minutes, held at ~200 °C for another 15 minutes, and cooled for 5 minutes. One filter background measurement was taken for every five cycles by keeping the filter on the desorption line. Raw data were saved every second and

were pre-averaged to a 10-second time interval before data processing. The data were analyzed using the data analysis toolkit Tofware (v2.5.11, Tofwerk, Thun, Switzerland and Aerodyne, Billerica, MA) within the Igor Pro software (v6.37, Wavemetrics, Portland, OR). The FIGAERO-CIMS particle data matrix was also generated from Tofware v2.5.11. The signals reported for particles in later discussion were integrations over the thermal desorption process, with background subtracted.  The signals are in

counts per second (Hz), if not specified in the following discussion. A uniform sensitivity was assumed for FIGAERO-CIMS measurements. Due to the nature of iodide reagent ion, which has a higher sensitivity towards oxygenated organic compounds (Lee et al., 2014), the importance of more oxidized compounds will be over-emphasized while less oxidized compounds under-emphasized. Nevertheless, a good correlation (R = 0.84) between total OA measured by AMS and FIGAERO-CIMS (Figure S1)

suggests that the assumption of uniform sensitivity to some extent could be reasonable in this study. When we compared the FIGAERO-CIMS measurements with AMS measurements, the FIGAERO-CIMS signals were converted to mass concentrations by multiplying ion signals in Hz with the molecular weight (MW) of each ion, and the new unit is $g\ mol^{-1}\ s^{-1}$. This conversion allows for an easier cross-instrument comparison between AMS and FIGAERO-CIMS.

This study focuses on AMS and FIGAERO-CIMS measurements. Other co-located instruments included PILS-ICs to measure water-soluble inorganic and organic acid species, CIMSs to measure gaseous species, PILS and mist chambers coupled to a total organic carbon (TOC) analyzer to measure particle- and gas-phase water-soluble organic carbon, and gas chromatography-flame ionization

detector (GC-FID) with a Markes focusing trap to measure hourly resolved VOC, and a chemiluminescence monitor to measure NO and $NO_2$.



## 2.3 Source apportionment methods

As organic measurements from the AMS and FIGAERO-CIMS are comprised of hundreds of species, source apportionment methods were applied to both measurements for a better understanding of OA sources and composition. Two widely used source apportionment methods, positive matrix factorization (PMF) and the multilinear engine (ME-2) algorithm, were used here. PMF is the most commonly used source apportionment method for AMS data (Lanz et al., 2007;Ulbrich et al., 2009;Jimenez et al., 2009;Ng et al., 2010;Zhang et al., 2011). It is a least-squares approach based on a receptor-only multivariate factor analytic model to solve bilinear unmixing problems. PMF deconvolves the observed data matrix as a linear combination of various factors with constant mass spectra but varying concentrations across the dataset. The model solution of PMF is not unique due to rotational ambiguity. The ME-2 solver works in a similar manner to PMF. The difference between PMF and ME-2 is that ME-2 allows users to introduce *a priori* information, in the form of a known factor time series and / or a factor profile, as inputs to the model to constrain the solution (Canonaco et al., 2013). In the following discussion, we applied PMF analysis to both AMS and FIGAERO-CIMS datasets, respectively. For the AMS dataset, we found that unconstrained PMF runs failed to identify reasonable solutions, so we performed ME-2 analysis on the AMS dataset and constrained it with a fixed Isoprene-OA factor profile. The constraining method was known as a-value approach (Canonaco et al., 2013;Crippa et al., 2014), where the a-value (ranging from 0 to 1) determines how much a factor profile is allowed to vary from the input source profile. The Isoprene-OA factor profile (anchor profile) we used to constrain the ME-2 analysis was previously resolved by PMF from Centreville, Alabama, during the SOAS campaign (Xu et al., 2015a;Xu et al., 2015b) . A description of our ME-2 analysis is provided in Section 3.

## 2.4 Estimating mass concentration of organic nitrate functionality from AMS measurements

The mass concentration of organic nitrate functionality ($NO_{3,org}$) was calculated based on $NO^+/NO_2^+$ from AMS measurements (Farmer et al., 2010), by eq. 1-2.

$$NO_{2,org} = \frac{NO_{2,meas} \times (R_{meas} - R_{AN})}{R_{ON} - R_{AN}} \qquad (1)$$

$$NO_{3,org} = NO_{2,org} \times (R_{ON} + 1) \qquad (2)$$



where $R_{meas}$ is the $NO^+/NO_2^+$ ratio from field measurements; $R_{AN}$ is the $NO^+/NO_2^+$ ratio of pure ammonium nitrate; and $R_{ON}$ is the $NO^+/NO_2^+$ ratio of pure organic nitrates. Note that $NO_{3,org}$ refers to the mass concentration of nitrate functionality only (-$ONO_2$). In this study, an $R_{AN}$ of 3 (average value from three IE calibrations of ammonium nitrate throughout the field measurements) was adopted for

$NO_{3,org}$ calculation. For $R_{ON}$, two values, an upper bound of 10 and a lower bound of 5, derived from β-pinene+$NO_3$· and isoprene+$NO_3$· systems, respectively, were adopted to acquire a $NO_{3,org}$ range for field measurements (Bruns et al., 2010;Boyd et al., 2015;Xu et al., 2015b).

## 3 Results and Discussion

### 3.1 Campaign overview and OA bulk properties

The meteorological data of the campaign have already been discussed in detailed in Nah et al. (2018a). Briefly, the two-month measurements were characterized by moderate temperature (average $24.0 \pm 4.0$ °C) and high RH (average $68.9 \pm 17.9$ %). Isoprene was the most abundant VOC (average $1.21 \pm 1.08$ ppb), followed by propane (average $0.84 \pm 0.39$ ppb), α-pinene (average $0.37 \pm 0.40$ ppb), and β-pinene (average $0.32 \pm 0.29$ ppb), making biogenic VOC the predominant OA precursors at Yorkville. A clear

decreasing trend was observed for isoprene concentration as temperature decreased throughout the campaign, which is consistent with the seasonal variation of isoprene emission (Seinfeld and Pandis, 2016). The Yorkville site is located in a rural environment with low but non-negligible $NO_x$ level, with an average NO and $NO_2$ concentrations of $0.15 \pm 0.35$ ppb and $2.2 \pm 1.8$ ppb, respectively.  NO was probably transported from roadways, peaking at around 10 am.

Organic species were the dominant component of NR-$PM_1$ (average $5.0 \pm 2.3$ μg m$^{-3}$), contributing 75 % to the total NR-$PM_1$ aerosol mass measured by AMS. The study mean diurnal trends of OA elemental ratios measured by both the AMS and FIGAERO-CIMS are shown in Figure 1. Since the nitrate functionality of organic nitrates largely fragments into $NO^+$ and $NO_2^+$ in the AMS (Farmer et al., 2010)

and will result in underestimated O:C and N:C values for OA, the nitrogen mass and oxygen mass from $NO_{3,org}$ have been added back in AMS O:C and N:C analysis. Compared to the OA measured by AMS,





the OA measured by FIGAERO-CIMS was more oxidized, with a lower H:C (by 0.08 compared to AMS H:C) and a higher O:C (by 0.17 compared to original AMS O:C, and by 0.10 compared to the upper bound of AMS O:C after including oxygen atoms from $NO_{3,org}$). This difference can be explained by the selective sensitivity of the iodide reagent ion, which has a higher sensitivity towards oxygenated

organic compounds (Lee et al., 2014). After including $NO_{3,org}$ in the AMS N:C calculation, the AMS N:C measurements fell into the range of the FIGAERO N:C measurements (average of 0.017 from FIGAERO; average of 0.006 to 0.025 from AMS). Both AMS and FIGAERO-CIMS measurements consistently showed O:C peaked in the afternoon while N:C peaked at night, suggesting that OA at Yorkville was more oxidized in the afternoon and organic nitrates accounted for a larger OA fraction

at night.

**3.2 Overview of organic compounds detected by FIGAERO-CIMS**

Figure 2(a) shows the normalized spectra (signals in mixing ratio) of FIGAERO-CIMS measurements. In total, 769 multifunctional organic compounds possessing 1 – 18 carbons have been identified in this study, of which 423 were CHO species (pOC, containing at least one carbon atom, at least one oxygen

atom, and an even number of hydrogen atoms), and 346 were nitrogen-containing CHON species that match the formula of a particulate organic nitrate (pON, containing one nitrogen atom, at least one carbon atom, three or more oxygen atoms, and an odd number of hydrogen atoms). Compounds not attached to an iodide ion were excluded, as their ionization mechanisms were uncertain. Organic nitrates containing two or more nitrogen atoms were not included in the discussion given they are much less

abundant compared to organic mononitrates. Since FIGAERO-CIMS cannot distinguish compounds of the same molecular formula but with different molecular structures, the detected organic nitrate compounds can be peroxy nitrates or multifunctional alkyl nitrates.

On average, pOC and pON contributed 87.7 ± 10.8 % and 12.3 ± 10.8 %, respectively, to total

FIGAERO-CIMS signals, while pOC and pON showed distinct diurnal patterns. pON had a higher contribution at night (Figure 2(b)), consistent with our observations of higher N:C at night, which was reported by previous FIGAERO-CIMS studies at other sites (Lee et al., 2016;Huang et al., 2019). The





pON fraction was also estimated using AMS nitrate measurements, where we calculated lower and upper bound of $NO_{3,org}$ using a $NO^+/NO_2^+$ ratio of 10 and 5, respectively, and then applied an average MW of 220 g mol$^{-1}$ (effective MW of all pON measured by FIGAERO-CIMS) to covert AMS $NO_{3,org}$ to mass concentration of organic nitrates (sum of mass of both organic and nitrate functionalities of the
organic nitrates). The resulting pON fraction (pON/(Org + $NO_{3,org}$), 5 – 18 %) was comparable to FIGAERO-CIMS measurements and also agreed with previous studies in the southeastern U.S. (Xu et al., 2015b;Ng et al., 2017). For a group of pON or pOC with the same carbon atom number, a bell-shaped distribution was observed as a function of oxygen atom number (Figure S2 and Figure S3), similar to observations from previous field measurements (Lee et al., 2016;Lee et al., 2018;Huang et
al., 2019).

The average effective formulae of pOC and pON are $C_{6.4}H_{9.0}O_{5.3}N_0$ and $C_{7.5}H_{11.6}O_{6.5}N_1$, respectively. A series of small organic compounds (MW < 80 g mol$^{-1}$) were detected by FIGAERO-CIMS in this study, some of which were in high abundance, e.g., $CH_2O_2$ and $C_2H_4O_3$. These ions should not be detected in
the particle phase due to expected high volatility and were likely thermal decomposition products of less-volatile molecules, not uncommon in FIGAERO thermograms (Stark et al., 2017;Schobesberger et al., 2018). The presence of these ions biased effective formulae and MWs calculations, thus the values reported in Table 1 could be smaller than the actual molecules. Meanwhile, these small but highly-oxidized fragments may also have a higher carbon oxidation state and bias the AMS elemental ratio
calculation as well. pON molecules on average had around one more carbon than pOC molecules, meaning pON was composed of larger molecules compared to pOC. In Figure 2, to better illustrate the difference between pOC and pON composition, we grouped pOC and pON species into four subgroups based on the carbon atom number, $C_{1-5}$, $C_{6-10}$, $C_{11-15}$, and $C_{>15}$. For both pOC and pON, compounds with fewer than 15 carbon atoms accounted for majority of total signals (99.8 ± 0.1 % for pOC and 99.6 ±
0.2 % for pON), with $C_{6-10}$ being the most dominant subgroup (53.4 ± 33.3 % in pOC and 65.8 ± 5.4 % in pON), followed by $C_{1-5}$ (42.4 ± 33.8 % in pOC and 26.9 ± 5.3 % in pON), and $C_{11-15}$ (4.0 ± 0.7 % in pOC and 7.0 ± 1.1 % in pON) (Figure 2(c) and (d)). pON contained a higher fraction from $C_{6-10}$ while pOC contained a higher fraction from $C_{1-5}$, explaining the difference in their average formulae. Each





subgroup showed distinct diurnal patterns, while the same subgroup exhibited similar trends in pOC and pON (Figure 2(e) and (f)). Specifically, $C_{1-5}$ species had a larger contribution during the daytime while $C_{6-10}$ species were more dominant during the night. This is consistent with emission of their potential precursors, where $C_{1-5}$ were more likely to arise from isoprene oxidation while $C_{6-10}$ were more

likely to arise from monoterpenes, though contributions from other sources, fragmentation of monoterpene products, and dimer formation in isoprene oxidation are also possible. There was a lack of a clear day-night contrast for $C_{11-15}$ species, likely due to their low concentrations, low instrument sensitivity, and/or formation from various sources.

**3.3 AMS OA factors**

We started our analysis with unconstrained PMF runs using the Solution Finder (SoFi 6.4) software. Three factors can be resolved by unconstrained runs, which are Isoprene-OA, LO-OOA, and MO-OOA. This three-factor solution was consistent with previous AMS measurements conducted in summer at Yorkville (Xu et al., 2015a;Xu et al., 2015b), in which no primary OA factor was resolved. However, the contribution from Isoprene-OA appeared to be largely overestimated in our unconstrained PMF

runs. The campaign-average Isoprene-OA fraction was $45 \pm 15$ % (Figure S4) and the fraction was as high as 90 % at the beginning of the campaign, when the emission of isoprene was higher. However, previous measurements at the same site showed that Isoprene-OA only accounted for 32.5 % of total OA in July (Xu et al., 2015a;Xu et al., 2015b). Meanwhile, the $f_{\text{C5H6O}}$ ($C_5H_6O^+$/OA, a tracer for isoprene-derived SOA (Hu et al., 2015)) of the resolved Isoprene-OA was 7.0 ‰ (Figure S4 (c)), while in

previous studies Isoprene-OA had an $f_{\text{C5H6O}}$ of around 20 ‰ (Hu et al., 2015;Xu et al., 2015b). These discrepancies indicated that the Isoprene-OA factor resolved by unconstrained PMF likely included interferences from other types of OA as measurements were conducted during transition in seasons (isoprene emissions), and that unconstrained PMF alone was not sufficient to identify the correct solution for this dataset. Therefore, we applied constraints in form of Isoprene-OA profile. In previous

studies, only the POA profile, rather than SOA, has been fixed in ME-2 analysis (Crippa et al., 2014;Elser et al., 2016). However, as Isoprene-OA is a commonly resolved biogenic SOA in the southeastern U.S. during summertime (Xu et al., 2015a;Xu et al., 2015b;Hu et al., 2015;Budisulistiorini



et al., 2016;Rattanavaraha et al., 2016) and its profile shows consistency in different studies (Hu et al., 2015), we constrained the Isoprene-OA profile with a "clean" Isoprene-OA profile resolved in the southeastern U.S. during summer 2013 SOAS measurements at Centreville (Xu et al., 2015a;Xu et al., 2015b). The rotations were explored using the a-value approach (Lanz et al., 2008;Canonaco et al., 2013;Crippa et al., 2014). We tested five a-values for the Isoprene-OA profile, from 0 to 0.8, with an increment of 0.2. The determination of a final solution was guided by three criteria: mass fraction of each factor (Figure S5(b)), correlation between factor time series with external tracers, and the $f_{C5H6O}$ of resolved Isoprene-OA (Figure S5(c)). Different external tracers were also used for identifying OA factors. 2-methyltetrol is the ring-opening product of IEPOX and can be measured by I$^-$-CIMS (Surratt et al., 2010;Lin et al., 2012;Hu et al., 2015). Lopez-Hilfiker et al. (2016) showed that the 2-methyltetrol signal detected in FIGAERO-CIMS may be derived from thermal decomposition of accretion products or other organics of lower volatility, but IEPOX uptake is still the major source for this fragment. Here, we still used the 2-methyltetrol ($C_5H_{12}O_4$) signal measured by FIGAERO-CIMS as a tracer species for Isoprene-OA. Xu et al. (2015a) showed that organic nitrates made up a substantial portion of LO-OOA in the southeastern U.S. and had a good correlation with LO-OOA. Thus, we used organic nitrate functionality as a tracer for LO-OOA.

Based on the above criteria, a three-factor solution with an a-value of 0 was chosen for the AMS dataset. The chosen three-factor solution gave the best correlations between Isoprene-OA and $C_5H_{12}O_4$ signal (R = 0.85), LO-OOA and $NO_{3,org}$ (R = 0.84), and the highest $f_{C5H6O}$ (23 ‰) (Figure S5). The mass spectra and time series for the factors are shown in Figure 3. With ME-2 analysis, the fraction of Isoprene-OA was lower compared to unconstrained PMF. On average, Isoprene-OA, LO-OOA, and MO-OOA contributed 17 ± 5 %, 33 ± 15 %, and 50 ± 13 % to total OA, respectively. Over the course of the campaign, the fraction of Isoprene-OA in total OA decreased from 26% to 8% (daily averages), consistent with the decreasing temperature during season transition (Figure S6). Similar to previous measurements at the same site (Xu et al., 2015a;2015b), MO-OOA was characterized by a wide afternoon peak, likely related to strong daytime photochemistry, while LO-OOA had a nighttime enhancement, which can arise from changes in boundary layer height, temperature-driven partitioning,



as well as nocturnal OA formation such as nitrate radical oxidation of biogenic VOC. The diurnal trend of Isoprene-OA also showed an afternoon enhancement, but the day-night contrast was less pronounced compared to MO-OOA. MO-OOA had the highest O:C (0.91), followed by Isoprene-OA (0.63) and LO-OOA (0.49).

## 3.4 FIGAERO-CIMS OA factors

The integration of each thermogram, with background subtracted, was taken as the total particle-phase signal (255 desorption cycles were measured in total). The factorization analysis was performed on the integrated total particle-phase signals in the Igor Pro based PMF Evaluation Tool (version 2.06). Initially, the errors of integrated signals were estimated using Poisson statistics as follows:

$$\sigma = \sqrt[2]{I} \qquad\qquad\qquad (3)$$

where $I$ is the integrated ion signal in the unit of ions. However, we noticed that the $\sigma$ values estimated by Poisson statistics only provide a lower limit for the real noise, probably due to unaccounted variabilities introduced by thermogram integration, which can be subjected to overlapping peaks and fragmented ions. As a consequence, the $Q/Q_{exp}$ from the PMF analyses is $>>1$ (Figure S7), indicating that the estimated errors were underrepresented (Ulbrich et al., 2009). Given the complexity of uncertainties associated with the thermal desorption processes and a lack of well-developed methods to estimate these uncertainties, we developed an empirical scaling factor by comparing the time series of several pairs of highly-correlated ions (Figure S8). Figure S8(a), for example, shows a scatter plot of two ions that are highly correlated as a function of time. The Poisson uncertainties for each data point, calculated according to eq. 3, are also shown. The measured scatter does not have any clear trend with time and is clearly much larger than the calculated Poisson uncertainties. Thus, the uncertainties input into the PMF analysis were empirically increased by a factor of 10 to better account for the observed scatter. This empirical scaling factor of 10 was applied to all errors, which gives more reasonable $Q/Q_{exp}$ values (Figure S9) and now only requires one factor to explain the highly-correlated ions. As discussed above, thermal decomposition processes could result in the production of a series of small organic compounds (MW < 80 g mol$^{-1}$). We included these small ions in the PMF analysis, since their time variations reflected those of their parent compounds, but including them will likely result in





overestimation of the carbon oxidation state and underestimation of the effective MWs of the factors in later discussion.

Carbon oxidation state of each FIGAERO-CIMS factor was calculated using a formula modified from that in Kroll et al. (2011) to include organic nitrate contributions, where a group oxidation state of -1 was applied to $-ONO_2$ functionality:

$$\overline{OS_c} = 2 \times (O:C - 3 \times N:C) - H:C + N:C \qquad (4)$$

which can be rewritten as

$$\overline{OS_c} = 2 \times O:C - H:C - 5 \times N:C \qquad (5)$$

As mentioned above, iodide reagent ion has a higher sensitivity towards oxygenated organic compounds. Meanwhile, the small and highly-oxidized organic compounds formed in potential thermal decomposition may have a higher carbon oxidation state than their parent molecules. Thus, the average carbon oxidation states calculated for FIGAERO-CIMS factors could be higher than the actual values.

Five FIGAERO-CIMS OA factors were resolved (Figure 4). Two factors showing clearly higher N:C (0.028 and 0.032) were distinguished by their diurnal trends and thus denoted as Day-ONRich (daytime ON-rich) factor and NGT-ONRich (nighttime ON-rich) factor. For the remaining three daytime factors with lower N:C (0.008, 0.009, and 0.011), one showed a significantly higher $\overline{OS_c}$ and was denoted as Day-MO (daytime more-oxidized, $\overline{OS_c} = 0.50$) factor, while the other two were distinguished by their diurnal trends and thus denoted as MRN-LO (morning less-oxidized) factor and AFTN-LO (afternoon less-oxidized) factor. Day-MO, Day-ONRich, MRN-LO, AFTN-LO, and NGT-ONRich factors accounted for 25 ± 15 %, 12 ± 10 %, 21 ± 13 %, 23 ± 16 %, and 18 ± 13 % of total signals measured by FIGAERO-CIMS, respectively. The average effective formulae and MWs were calculated for each factor, as well as for their pOC and pON components, and are shown in Table 1. Similar to the discussion in Section 3.2, the pOC and pON species of each factor were grouped into and discussed as $C_{1-5}$, $C_{6-10}$, $C_{11-15}$, and $C_{>15}$ subgroups (Figure 5). The concentration of $C_{>15}$ subgroup was negligible, so we excluded them from the following discussion. Below, we evaluate and discuss tracer ions for each



FIGAERO-CIMS OA factor, based on both their absolute abundance (i.e., ions of the highest signal in the mass spectrum of each factor) and their fractional abundance (i.e., ions dominantly presented in a certain factor).

NGT-ONRich had the largest MW (193.4 g mol$^{-1}$), highest effective carbon atom number (7.0), and lowest $\overline{OS_c}$ (0.13), meaning this factor was composed of larger and less oxidized molecules. This feature can be seen more clearly in Figure 5. Compared to the other four factors, both pOC and pON of NGT-ONRich had a larger fraction from $C_{6-10}$ and $C_{11-15}$ subgroups, and a smaller fraction from $C_{1-5}$ subgroup. NGT-ONRich also had the highest effective nitrogen atom number (0.22), meaning one in every five

molecules was an organic nitrate. The most abundant pON species in NGT-ONRich were $C_5H_9NO_7$ and $C_{10}H_{15}NO_8$, accounting for 7.8 and 3.5 % of pON signals in this factor, respectively. $C_{10}H_{15}NO_8$ has been characterized in multiple chamber studies as major products of α-/β-pinene/limonene+NO$_3$· and α-/β-pinene photooxidation with the presence of NO$_x$ (Nah et al., 2016;Lee et al., 2016;Faxon et al., 2018;Takeuchi and Ng, 2019). At Yorkville, the majority of $C_{10}H_{15}NO_8$ was presented in NGT-ONRich,

implying that nocturnal chemistry is its most important source. Besides $C_{10}H_{15}NO_8$, a series of $C_{9,10}$ pON ($C_9H_{9,11,13}NO_{8,9,10}$ and $C_{10}H_{13,15,17}NO_{8,9,10}$) were also dominantly presented in NGT-ONRich, which were similar to fingerprint ions reported by Massoli et al. (2018) for gaseous terpene nitrate factor at Centreville during the SOAS campaign. The NGT-ONRich we resolved here is likely the particle-phase counterpart of that gaseous terpene nitrate factor. $C_5H_9NO_7$ was not solely present in NGT-ONRich.

Instead, it contributed an even higher fraction to Day-ONRich, suggesting that both daytime and nighttime pathways were critical for $C_5H_9NO_7$ at Yorkville. This is consistent with $C_5H_9NO_7$ being detected in previous laboratory studies on isoprene+NO$_3$· and isoprene photooxidation in the presence of NO$_x$ (Ng et al., 2008;Lee et al., 2016). Both $C_5H_9NO_7$ and $C_{10}H_{15}NO_8$ have also been identified at Centreville in rural Alabama, U.S., during SOAS, among the top ten most abundant pON species (Lee

et al., 2016). In another field study at the boreal forest research station SMEAR II located in Hyytiälä, southern Finland, $C_{10}H_{15}NO_8$ has been suggested to be a fingerprint molecule for a daytime factor measured with NO$_3^-$-based CI-APi-TOF (Yan et al., 2016), but in this study it was more abundant at night. The pOC tracer of NGT-ONRich was $C_8H_{12}O_5$, likely corresponding to 2-hydroxyterpenylic acid,



which was proposed to be an α-pinene SOA tracer formed from further oxidation of terpenylic acid (Eddingsaas et al., 2012;Kahnt et al., 2014a;Kahnt et al., 2014b;Sato et al., 2016). Taken together, the high contribution from $C_{6-10}$ subgroup and the presence of quite a few monoterpene SOA tracers in NGT-ONRich strongly related this factor to monoterpene chemistry, with a non-negligible contribution

from isoprene organic nitrates. NGT-ONRich also contained the highest fraction of $C_{11-15}$ group. While most signals were from $C_{11}$ ions, we also observed some $C_{14}$ and $C_{15}$ compounds, e.g., pOC $C_{14}H_{18-22}O_{5-7}$ and $C_{15}H_{20-24}O_{5-7}$, pON $C_{14}H_{21-25}NO_7$ and $C_{15}H_{23-27}NO_7$, which possibly originated from sesquiterpene oxidation, though more fundamental laboratory studies are needed to further constrain this.

Day-ONRich had an effective nitrogen atom number of 0.16, lower compared to NGT-ONRich, but still significantly higher than other daytime factors. 23 % of Day-ONRich pON signals was from $C_5H_9NO_7$, implying isoprene as the crucial precursor of Day-ONRich, even considering half of $C_5H_9NO_7$ signal may arise from fragmentation of other larger molecules (Figure S10(a)). The second highest pON ion, $C_5H_7NO_7$, was also likely from isoprene. The high signals from $C_5H_7NO_7$ and

$C_5H_9NO_7$ made the $C_{1-5}$ ON subgroup as prevalent as the $C_{6-10}$ ON subgroup, which was a distinctive feature for Day-ONRich (Figure 5). Meanwhile, the pOC of Day-ONRich also contained noticeably more $C_{1-5}$ ions than other factors, probably due to fragmentation process being a favored pathway under high-NO conditions (Kroll and Seinfeld, 2008). As a result, Day-ONRich had the lowest effective MW (164.5 g mol$^{-1}$) and the lowest effective carbon number (5.6). The most abundant pOC species of Day-

ONRich were $C_3H_4O_5$, $C_4H_6O_5$, and $C_5H_8O_5$. The formula of $C_3H_4O_5$ implied dicarboxylic acid and it has been reported in aqueous processes (Lim et al., 2010). However, the average thermogram of $C_3H_4O_5$ showed two peaks (Figure S10(b)), where the first peak ($T_{max}$ = 74.3 °C) roughly matched the volatility of $C_3$ dicarboxylic acids and the second peak ($T_{max}$ = 113.2 °C) likely came from thermal decomposition of molecules of lower volatility. Similar multiple-peak behavior was observed for $C_3H_4O_4$, a tracer

compound for Day-MO (Figure S10(c)). $C_4H_6O_5$, possibly malic acid, has been reported as a higher-generation product of unsaturated fatty acids photochemistry (Kawamura et al., 1996), but has also been found in isoprene SOA in several studies, including particle-phase reactions in isoprene photooxidation in the presence of NO$_x$, non-IEPOX pathway via ISOPOOH+OH· reaction (ISOPOOH-SOA), and





isoprene ozonolysis (Nguyen et al., 2010;Xu et al., 2014;Krechmer et al., 2015). One isomer of $C_5H_8O_5$, 3-hydroxyglutaric acid, has been used as a tracer for α-/β-pinene photooxidation SOA (Claeys et al., 2007), while other studies have identified $C_5H_8O_5$ in isoprene SOA when the IEPOX pathway was suppressed (Nguyen et al., 2011;Krechmer et al., 2015;Liu et al., 2016). $C_5H_8O_5$ was also found in the

oxidation of 1,3,5-trimethylbenzene (Praplan et al., 2014), toluene (Kleindienst et al., 2007), and levoglucosan (Zhao et al., 2014). There was no sign of prevalent anthropogenic emissions or biomass burning events during the measurements, so the presence of $C_5H_8O_5$ was more likely linked to monoterpene photooxidation and/or non-IEPOX isoprene chemistry.

Day-MO was dominated by pOC signals (accounting for 95 % of signals) and characterized by the highest $\overline{OS_c}$ (0.50) of all factors. The tracer ions of Day-MO were $C_4H_4O_6$, $C_5H_6O_6$, and $C_5H_8O_6$. Given their lower degree of saturation and considerably high O:C, these compounds were likely carboxylic acids, particularly di- or even tri-carboxylic acids. For instances, $C_4H_4O_6$, likely 2-hydroxy-3-oxosuccinic acid, was identified in OH· initiated oxidation of aqueous succinic and tartaric acids (Chan

et al., 2014;Cheng et al., 2016). $C_5H_8O_6$ was likely 2,3-dihydroxy-2-methylsuccinic acid, a product of aqueous cross photoreaction of glycolic and pyruvic acids (Xia et al., 2018), or methyltartaric acids (MTA), tracers of aged isoprene SOA (Jaoui et al., 2019). However, we cannot rule out the possibility that they were fragments from thermal decomposition of larger molecules. Techniques without thermal desorption processes will be beneficial in understanding the nature of highly-oxidized OA molecules in

future studies.

Similar to Day-MO, pOC accounted for more than 90% of total signals in MRN-LO and AFTN-LO. These two factors had similar fractions from each subgroup (Figure 5), though they were dominated by different ions. For MRN-LO, the dominating ions were $C_8H_{12}O_5$ and $C_3H_4O_4$, while $C_7H_{10}O_5$ also stood

out. $C_8H_{12}O_5$, as discussed above, was related to α-/β-pinene SOA, and $C_7H_{10}O_5$ also likely corresponded to an α-pinene SOA tracer, i.e., 3-acetylpentanedioic acid (Kleindienst et al., 2007). $C_3H_4O_4$ could correspond to malonic acid or its isomers, but given its high desorption temperature (Figure S10(c)), $C_3H_4O_4$ was more likely fragments of larger molecules. For AFTN-LO, the most



prominent ions were $C_4H_4O_6$, $C_5H_{10}O_{4,5}$, and $C_9H_{14}O_{4,5}$. $C_4H_4O_6$, as discussed above, was likely related to aqueous processing. $C_9H_{14}O_4$, likely pinic acid (Seinfeld and Pandis, 2016), was a well-established fresh α-pinene SOA tracer, and $C_9H_{14}O_5$ was probably related to α-/β-pinene SOA (Kahnt et al., 2014a;Kahnt et al., 2014b;Sato et al., 2016). $C_5H_{10}O_5$ has been shown to be a dominant product of

ISOPOOH-SOA (Krechmer et al., 2015;D'Ambro et al., 2017), but has also been detected in isoprene ozonolysis and isoprene photooxidation under high-NO (Jaoui et al., 2019). It is interesting that a non-IEPOX isoprene SOA product was found to be one of the prominent tracers for an afternoon low-NO fresh SOA factor in our study. Previous factorization analysis of AMS measurements alone suggested that ISOPOOH-SOA accounted for only ~2 % of ambient OA at Centreville during summer 2013 SOAS

measurements (Krechmer et al., 2015). If the $C_5H_{10}O_5$ we observed in AFTN-LO was dominantly from ISOPOOH+OH· reaction via non-IEPOX pathway, ISOPOOH-SOA may account for a more considerable fraction of fresh isoprene SOA in our study compared to that reported in Centreville. Thus, the initial difficulty we encountered when resolving Isoprene-OA, which is believed to form mainly via the IEPOX pathway, from PMF analysis of AMS data may be explained to some extent. Taken together,

although both MRN-LO and AFTN-LO were relatively fresh SOA, MRN-LO had more contribution from monoterpenes, while AFTN-LO was more dominated by isoprene SOA.

### 3.5 Tracer species detected by FIGAERO-CIMS and their implications

As discussed in Section 3.4, a series of biogenic SOA tracers, mostly from isoprene and monoterpenes, has established their importance in more than one FIGAERO-CIMS OA factor. To better understand

the OA formation mechanisms, we selected six isoprene and monoterpene SOA tracers to represent different oxidation pathways and examined their distributions in the five FIGAERO-CIMS OA factors (Figure 6).

For isoprene SOA, $C_5H_9NO_7$ was chosen here as pON tracer, $C_5H_{12}O_4$ as IEPOX uptake tracer, and

$C_5H_{10}O_5$ as non-IEPOX tracer. Note that $C_5H_{10}O_5$ can form from isoprene oxidation under various conditions: while $C_5H_{10}O_5$ is a major product in ISOPOOH+OH· when the IEPOX uptake pathway is suppressed (Krechmer et al., 2015;D'Ambro et al., 2017), it also forms in isoprene+$O_3$ and isoprene+





OH·+$NO_x$ (Jaoui et al., 2019). Most of the $C_5H_9NO_7$ signals were found in Day-ONRich (39 %) and NGT-ONRich (32 %), suggesting a non-negligible isoprene ON formation during both day and night. The efficient nocturnal isoprene oxidation is possibly via the reaction with nitrate radicals rather than with ozone (Ng et al., 2008;Brown et al., 2009;Schwantes et al., 2015;Fry et al., 2018). In addition, the

recent work by Fry et al. (2018) suggested a substantially longer nighttime peroxy radical lifetime in ambient air versus under chamber conditions, which allows for the formation of lower-volatility products and thus higher SOA yields from isoprene nocturnal chemistry. $C_5H_{12}O_4$ was only noticeable in daytime, non-ON-Rich factors, consistent with its low-NO photochemistry origin. $C_5H_{10}O_5$ was also only present in daytime factors. However, different from $C_5H_{12}O_4$, a noticeable fraction of its signal

was in Day-ONRich, implying that $C_5H_{10}O_5$ can also be formed under high-NO conditions. One interesting observation was that while $C_5H_{12}O_4$ is an early-generation product of isoprene oxidation, it had a larger fraction in Day-MO (expected to be aged SOA) than in AFTN-LO (expected to be fresh SOA). Here, we hypothesize that the Day-MO factor was closely related to particle-phase aqueous processes, and the presence of $C_5H_{12}O_4$ in Day-MO can be explained by that IEPOX uptake to the

particle phase requires aerosol water. Aqueous chemistry can also explain the acid-like ions observed in large abundance in Day-MO.

For monoterpene SOA, $C_{10}H_{15}NO_8$ was used here as pON tracer, $C_9H_{14}O_4$, likely pinic acid (Seinfeld and Pandis, 2016), as fresh SOA tracer, and $C_8H_{12}O_6$, likely 3-methyl-1,2,3-butanetricarboxylic acid

(MBTCA) (Müller et al., 2012;Eddingsaas et al., 2012), as an aged SOA tracer. $C_{10}H_{15}NO_8$ was prominently present in the nighttime factor NGT-ONRich, implying that nocturnal oxidation, likely by nitrate radicals, was its major source. The majority of $C_9H_{14}O_4$ signal was found in MRN-LO and AFTN-LO as expected, consolidating MRN-LO and AFTN-LO as daytime fresh SOA factors. $C_8H_{12}O_6$ was suggested to form from OH-initiated oxidation of pinonic acid in the gas phase (Müller et al., 2012),

but at Yorkville it was present in comparable abundance in MRN-LO, AFTN-LO, Day-MO, and NGT-ONRich, suggesting that complex aging pathways of fresh monoterpene SOA took place both day and night.





### 3.6 Correlations between AMS OA factors and FIGAERO-CIMS OA factors

To compare AMS OA factors with FIGAERO-CIMS OA factors, we first converted FIGAERO-CIMS signals (Hz) to mass concentration (Hz g mol$^{-1}$) by simply applying the effective MW to the time series of each factor, while still assuming uniform sensitivity for all compounds. The hourly averages were

used for cross-instrument comparison and results are shown in Figure 7.

For both AMS and FIGAERO-CIMS measurements, only one nighttime factor was resolved, LO-OOA from AMS and NGT-ONRich from FIGAERO-CIMS. A good correlation (R = 0.77) in time series was observed between them (Figure 7(c) and (d)). As discussed above, the FIGAERO-CIMS measurements

strongly related this factor to monoterpene chemistry, which was consistent with previous AMS measurements in the southeastern U.S. (Xu et al., 2015a;Xu et al., 2015b). NGT-ONRich also showed a prevalence contribution from organic nitrates, with one fourth of molecules being pON species. However, FIGAERO-CIMS also identified a non-negligible presence of isoprene-derived pON species in this factor, which the AMS was unable to resolve, implying the potential contribution from isoprene

nocturnal organic nitrate formation. In a recent study, Xu et al. (2018) showed that the major source of LO-OOA in the southeastern U.S. is from monoterpenes, but also includes contributions from sesquiterpene oxidation pathways. Our observation of a series of $C_{14}$ and $C_{15}$ species in NGT-ONRich is consistent with the presence of sesquiterpene SOA, though it cannot provide a further quantitative constraint.

Two daytime factors were resolved for AMS measurements, while four were resolved for FIGAERO-CIMS measurements. Strong correlation was observed for the summation of the AMS daytime factors (Isoprene-OA + MO-OOA) and the summation of the FIGAERO-CIMS daytime factors (Day-MO + Day-ONRich + MRN-LO + AFTN-LO), with R = 0.89 (Figure 7(a) and (b)). For daytime factors, the

Day-ONRich factor was unique to FIGAERO-CIMS. In the AMS, the nitrate functionalities of pON fragmented into $NO^+$ and $NO_2^+$ ions, which were not included in source apportionment analysis, and may explain the difficulty of resolving daytime ON-rich factors for AMS dataset. Both AMS and FIGAERO-CIMS resolved one daytime aged SOA factor, i.e., AMS MO-OOA factor and FIGAERO-





CIMS Day-MO factor, and these two factors were well correlated (R = 0.71). For AMS MO-OOA, different theories regarding its sources and formation pathways have been proposed (which are not mutually exclusive), including photochemical aging of fresh OA (Jimenez et al., 2009;Ng et al., 2010;Bougiatioti et al., 2014), aqueous processes (Xu et al., 2017), formation of highly oxygenated

molecules (HOMs) (Ehn et al., 2014), long-range transport (Hayes et al., 2013), and entrainment of aged SOA from the residual layer (Nagori et al., 2019). In our previous discussion, we tentatively related FIGAERO-CIMS Day-MO, which correlated with AMS MO-OOA, to aqueous processes, but cannot rule out other processes. AMS resolved only one daytime fresh SOA factor, Isoprene-OA. Isoprene-OA was largely, but not entirely, attributed to IEPOX uptake (Xu et al., 2015a;Schwantes et al., 2015), and

the enhanced signal at *m/z* 82 ($C_5H_6O^+$) may arise from methylfuran-like structures (Robinson et al., 2011;Budisulistiorini et al., 2013;Hu et al., 2015). FIGAERO-CIMS resolved two daytime fresh SOA factors, MRN-LO and AFTN-LO. The summation of MRN-LO and AFTN-LO showed good correlation with AMS isoprene-OA factor (R = 0.76). We observed various ions with high abundance in MRN-LO and AFTN-LO that were likely associated with isoprene organic nitrates, isoprene oxidation via non-

IEPOX pathways, and monoterpene oxidation. Previous studies have shown that IEPOX-SOA was enhanced even under high-NO conditions (Jacobs et al., 2014;Schwantes et al., 2019) and that α-pinene SOA could interfere with AMS Isoprene-OA apportionment (Xu et al., 2018). All these observations may suggest a more complex origin for the AMS Isoprene-OA factor (i.e., not just IEPOX uptake).

### 3.7 Change of the abundance of biogenic VOC and AMS OA factors in a transitional period

This field campaign took place during the transition in seasons from summer to fall, where decreasing temperature led to changes in abundances of SOA precursors. Figure 8 shows the mixing ratios of major VOC (isoprene, α-pinene, and β-pinene) and mass concentrations of AMS OA factors as a function of temperature. The FIGAERO-CIMS factors were not discussed here because fewer data points were measured by FIGAERO-CIMS and were not sufficient to provide statistically reliable results. To

eliminate the influence of daily meteorological variations, two sampling periods with relatively stable meteorological conditions were chosen to represent daytime (12:00 – 16:00, high temperature and boundary layer height, peak solar radiation) and nighttime (00:00 – 04:00, low temperature and





boundary layer height, zero solar radiation), respectively. Isoprene mixing ratio showed a strong dependence on temperature in both day and night. The mixing ratios of α-pinene and β-pinene were moderately dependent on temperature when temperature was lower than 25 °C, and remained relatively constant when the temperature was higher than 25 °C, where most daytime data points resided. For

AMS factors, Isoprene-OA increased with temperature, followed the trend of isoprene, as expected. Meanwhile, different from isoprene, for the same temperature bin, the nighttime Isoprene-OA concentration was always higher than daytime concentration. This can be explained by that the high concentration of nighttime Isoprene-OA was also residue from daytime formation, but its concentration decreased with a slower rate given the longer lifetime of aerosol compared to gas species. The strong

dependence of Isoprene-OA on temperature suggested isoprene as the dominant precursor of this factor, implying that Isoprene-OA resolved from AMS measurements is still a good surrogate of isoprene-derived SOA even with the potential interference from monoterpene SOA as discussed above. LO-OOA showed similar trends to monoterpenes, consistent with our discussion above and previous literature that monoterpenes are the dominant precursors to LO-OOA in this region. For MO-OOA, a mild

dependence on temperature was observed, suggesting that at least some of its sources were affected by temperature, e.g., through aging of isoprene-derived SOA (emission of isoprene is temperature dependent).

## 4 Conclusions

Two-months of measurements were performed at a rural site in the southeastern U.S. during a transition

in seasons. AMS and FIGAERO-CIMS measurements were combined to provide a better understanding of OA sources, composition, and properties. Both instruments consistently identified more oxidized OA in the afternoon and enhanced pON formation during the night, although the OA measured by FIGAERO-CIMS was more oxidized than that by AMS, due to the nature of iodide reagent ion that was used in FIGAERO-CIMS. Similar AMS OA factors were resolved compared to previous summer

measurements at the same site, which were Isoprene-OA, LO-OOA, and MO-OOA (and no HOA). The fraction of AMS Isoprene-OA in total OA decreased from 26 % to 8 % over the campaign, concurrent





with decreasing isoprene mixing ratio, which was strongly dependent on temperature. For FIGAERO-CIMS, three daytime fresh OA factors with low N:C (MRN-LO, AFTN-LO, and Day-MO) each accounted for about one fourth of total signals measured by FIGAERO-CIMS, and two factors with high N:C (Day-ONRich and NGT-ONRich) together accounted for the rest. MRN-LO and AFTN-LO

were likely fresh biogenic SOA, with MRN-LO more dominated by monoterpene SOA and AFTN-LO more dominated by isoprene SOA. Day-MO was hypothesized to be a mixture of aged and fresh SOA whose formation was possibly aided by aerosol water. NGT-ONRich was mostly from nocturnal monoterpene chemistry, while daytime isoprene oxidation under the effects of $NO_x$ was more important to Day-ONRich. Lastly, a series of $C_{14}$ and $C_{15}$ compounds were identified by FIGAERO-CIMS,

possibly originated from sesquiterpene oxidation pathways. In this study, a uniform sensitivity was assumed for all species measured by FIGAERO-CIMS, resulting in some uncertainties in the overall elemental ratios and carbon numbers. Future studies are warranted to continue to characterize and optimize instrument sensitivity for further quantitative analysis.

Previous studies (Qi et al., 2019;Stefenelli et al., 2019) have shown that combinations of AMS and molecular based mass spectrometric information is a way forward to provide more insights into the nature of SOA in general. In this study, factor analysis of FIGAERO-CIMS data provided new insights into the sources and composition of the typical AMS OA factors observed in the southeastern U.S. Specifically, while the AMS Isoprene-OA factor has been largely attributed to IEPOX uptake in

previous studies, we identified more pathways of isoprene oxidation that contributed to isoprene SOA formation in addition to IEPOX uptake. Notable isoprene pON formation was observed, likely from photooxidation in the presence of $NO_x$ and nitrate radical oxidation, as well as notable ISOPOOH-SOA (ISOPOOH oxidation products via non-IEPOX pathways); both pathways have not been resolved by AMS analysis before. AMS LO-OOA factor correlated well with NGT-ONRich factor resolved by

FIGAERO-CIMS, which contained a series of monoterpene SOA tracers, consolidating that LO-OOA was mostly attributed to monoterpene SOA in the southeastern U.S. Nonetheless, the non-negligible isoprene-derived pON in NGT-ONRich factor also related it to nocturnal isoprene chemistry, which was not identified by previous AMS factorization analysis.



*Data Availability*. Data are available upon request from the corresponding author (ng@chbe.gatech.edu).

*Author Contributions*. Y.C., R.J.W., and N.L.N. designed research. Y.C., T.N., and K.B. performed research. Y.C. and M.T. processed AMS and FIGAERO-CIMS data. L.X., M.R.C., H.S., F.C., and A.S.H.P. provided insights for source apportionment analysis. Y.C. and N.L.N. analyzed data and wrote the paper.

*Competing Interests*. The authors declare no conflict of interest.

*Acknowledgements*. This work was supported by U.S. Environmental Protection Agency STAR grant R835882. It has not been formally reviewed by the EPA. The views expressed in this document are solely those of the authors and do not necessarily reflect those of the EPA. The EPA does not endorse any products or commercial services mentioned in this publication. M.T. and L.X. acknowledged support from NSF CAREER AGS-1555034. The FIGAERO-HR-ToF-CIMS was purchased through NSF Major Research Instrumentation (MRI) Grant 1428738. The authors want to thank Eric S. Edgerton for providing SEARCH network measurements and meteorological data, Qian Zhang, James Rowe, and Linghan Zeng for their help during the campaign.





**Table 1 Effective Molecular Composition of FIGAERO Factors**

| | Effective Formula | Effective MW (g/mol) | O:C | H:C | N:C | $\overline{OS_C}$ | Marker Ions |
|---|---|---|---|---|---|---|---|
| **Day-MO** | $C_{6.1}H_{8.1}O_{5.7}N_{0.05}$ | 173.0 | 0.94 | 1.33 | 0.009 | 0.50 | |
| **Day-MO (pOC)** | $C_{6.0}H_{8.0}O_{5.7}N_0$ | 171.4 | 0.94 | 1.33 | 0 | 0.56 | $C_4H_4O_6$, $C_5H_6O_6$, $C_5H_8O_6$ |
| **Day-MO (pON)** | $C_{6.9}H_{9.8}O_{6.0}N_1$ | 203.0 | 0.87 | 1.41 | 0.14 | -0.39 | |
| **Day-ONRich** | $C_{5.6}H_{8.1}O_{5.4}N_{0.16}$ | 164.5 | 0.96 | 1.43 | 0.028 | 0.35 | |
| **Day-ONRich (pOC)** | $C_{5.5}H_{7.6}O_{5.1}N_0$ | 154.8 | 0.94 | 1.40 | 0 | 0.47 | $C_3H_4O_5$, $C_4H_6O_5$, $C_5H_8O_5$ |
| **Day-ONRich (pON)** | $C_{6.7}H_{10.4}O_{7.0}N_1$ | 216.7 | 1.05 | 1.56 | 0.15 | -0.22 | $C_5H_9NO_7$, $C_5H_7NO_7$ |
| **MRN-LO** | $C_{6.6}H_{9.3}O_{5.2}N_{0.06}$ | 172.2 | 0.79 | 1.41 | 0.008 | -0.13 | |
| **MRN-LO (pOC)** | $C_{6.5}H_{9.1}O_{5.2}N_0$ | 170.2 | 0.80 | 1.40 | 0 | 0.19 | $C_8H_{12}O_5$, $C_3H_4O_4$, $C_7H_{10}O_5$ |
| **MRN-LO (pON)** | $C_{7.6}H_{11.7}O_{5.7}N_1$ | 207.0 | 0.75 | 1.55 | 0.13 | -0.71 | |
| **AFTN-LO** | $C_{6.7}H_{10.1}O_{5.4}N_{0.07}$ | 177.7 | 0.79 | 1.49 | 0.011 | 0.04 | |
| **AFTN-LO (pOC)** | $C_{6.7}H_{9.8}O_{5.3}N_0$ | 174.5 | 0.80 | 1.48 | 0 | 0.12 | $C_4H_4O_6$, $C_5H_{10}O_5$, $C_5H_{10}O_4$, $C_9H_{14}O_4$, $C_9H_{14}O_5$ |
| **AFTN-LO (pON)** | $C_{7.8}H_{13.0}O_{6.0}N_1$ | 217.7 | 0.77 | 1.66 | 0.13 | -0.76 | |
| **NGT-ONRich** | $C_{7.0}H_{10.0}O_{6.0}N_{0.22}$ | 193.4 | 0.85 | 1.41 | 0.032 | 0.13 | |
| **NGT-ONRich (pOC)** | $C_{6.9}H_{9.5}O_{5.7}N_0$ | 182.9 | 0.83 | 1.38 | 0 | 0.28 | $C_8H_{12}O_5$ |
| **NGT-ONRich (pON)** | $C_{7.7}H_{11.7}O_{7.0}N_1$ | 230.0 | 0.91 | 1.51 | 0.13 | -0.35 | $C_5H_9NO_7$, $C_{10}H_{15}NO_8$ |

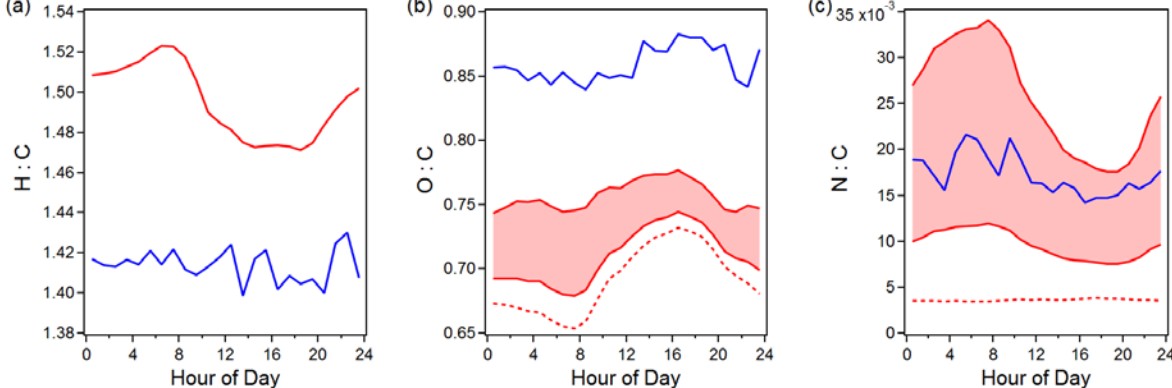

**Figure 1 Study mean diurnal trends of elemental ratios measured by AMS (red) and FIGAERO-CIMS (blue). The AMS O:C and N:C with and without including NO$_{3,org}$ are in shaded area (with NO$^+$/NO$_2^+$ ratio of 5 and 10) and in dashed line, respectively.**



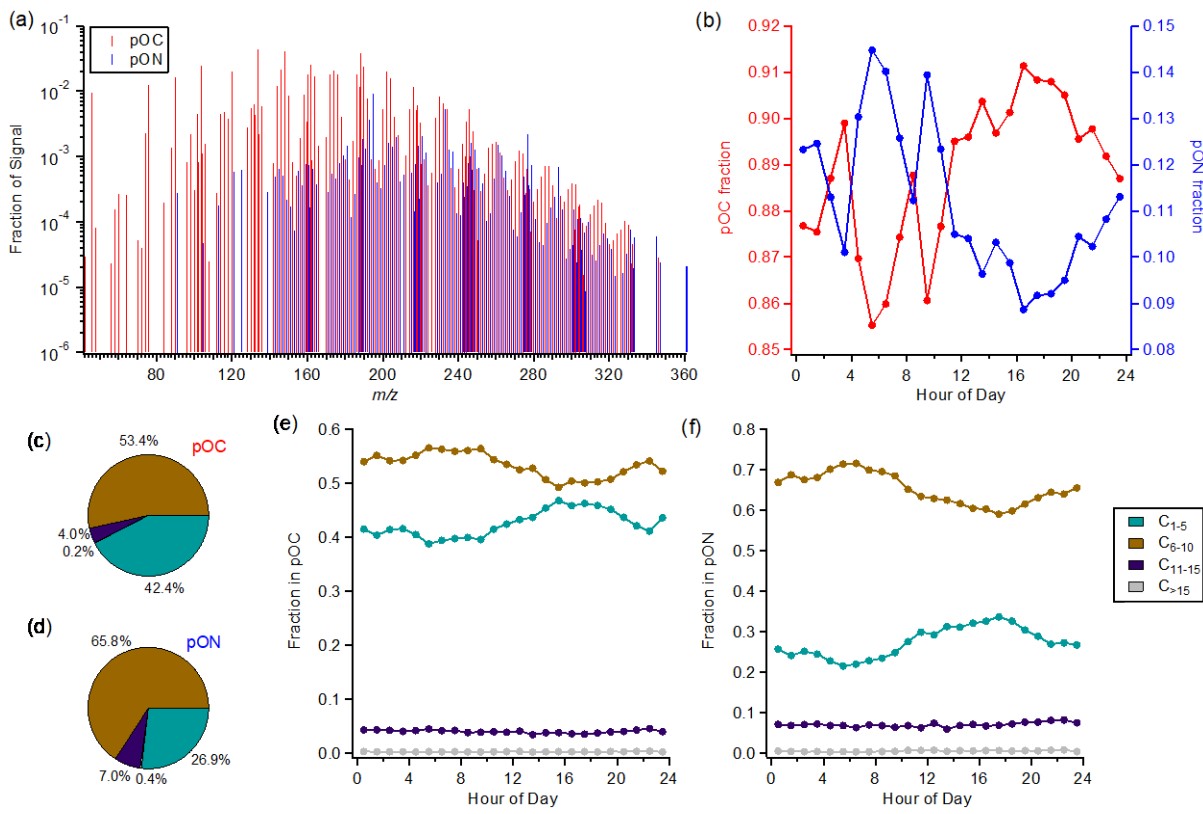

**Figure 2** Study mean (a) FIGAERO mass spectra ($C_xH_yO_z$ ions in red and $C_zH_yO_zN_1$ ions in blue), (b) fraction of pOC and pON compounds plotted as a function of time of a day, (c) and (d) fraction of ions of different carbon numbers (grouped as $C_{1-5}$, $C_{6-10}$, $C_{11-15}$, and $C_{>15}$) in pOC and pON, and (e) and (f) fraction of $C_{1-5}$, $C_{6-10}$, $C_{11-15}$, and $C_{>15}$ compounds in pOC and pON plotted as a function of time of day.





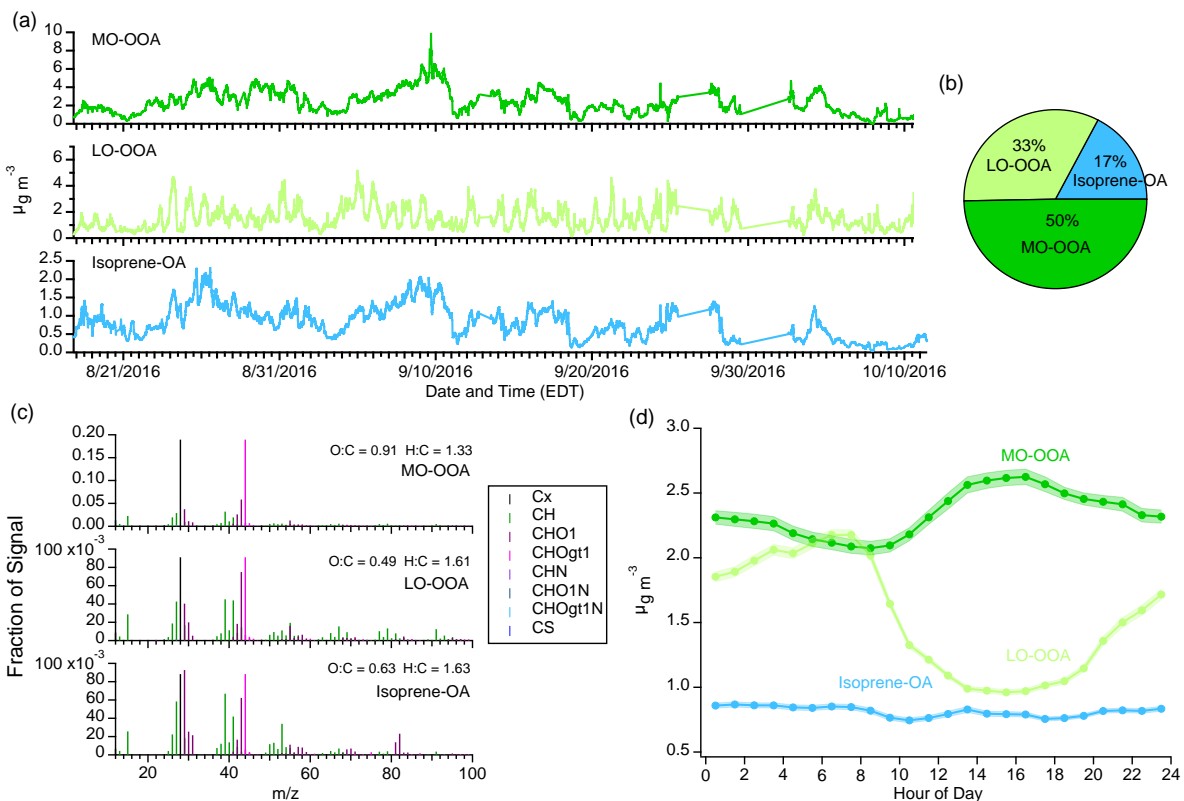

**Figure 3 (a) Time Series, and study mean (b) mass fraction, (c) normalized mass spectra, and (d) diurnal profiles (standard deviations in shaded areas) of AMS OA factors resolved by ME-2.**





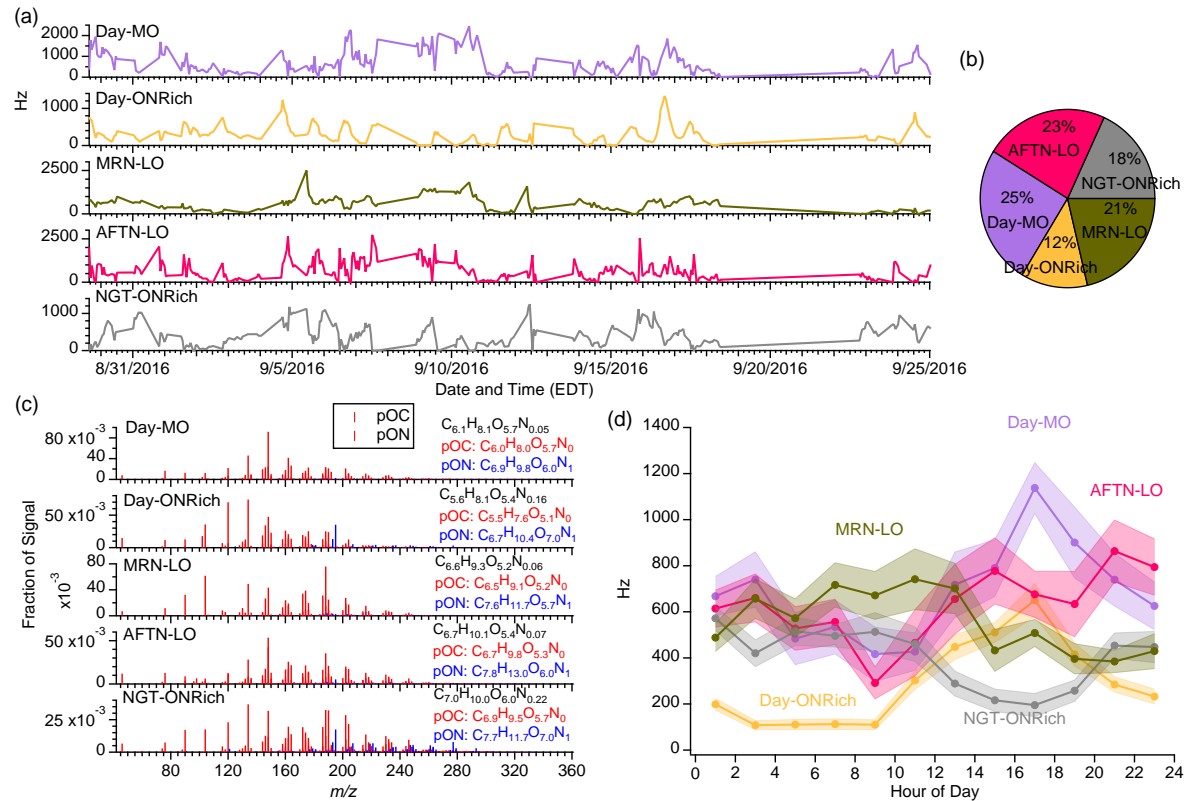

**Figure 4 (a) Time series, and study mean (b) fraction, (c) normalized mass spectra, and (d) diurnal profiles (standard deviations in shaded areas) of FIGAERO OA factors resolved by PMF.**



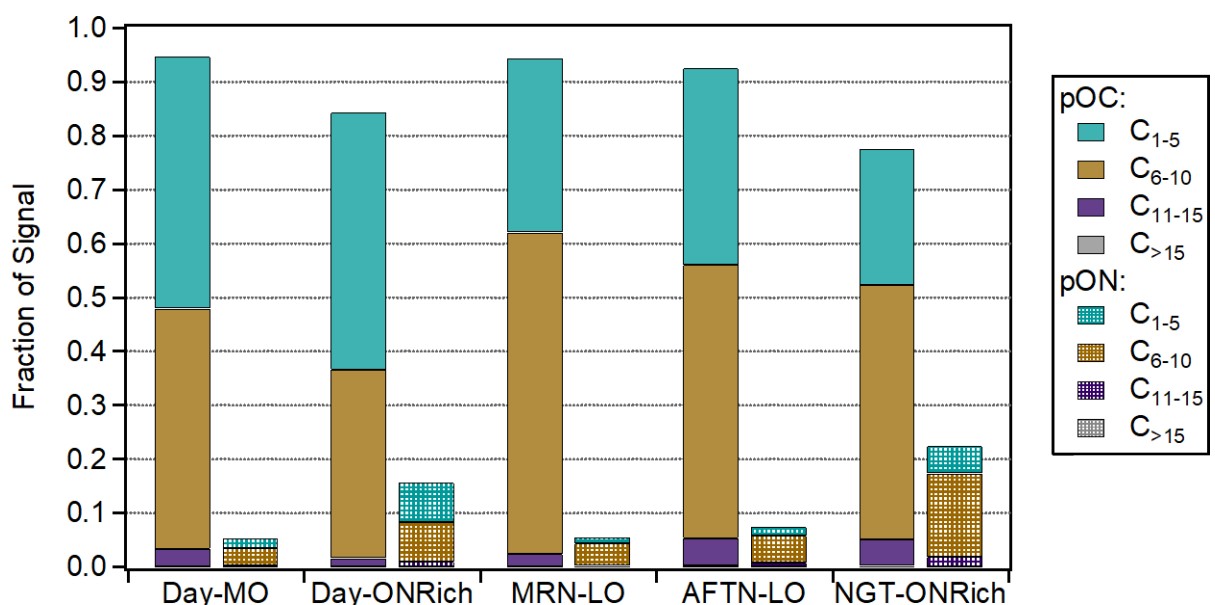

**Figure 5 Fraction of pOC and pON ions of different carbon numbers (grouped as $C_{1-5}$, $C_{6-10}$, $C_{11-15}$, and**
5  **$C_{>15}$) in each FIGAERO OA factor.**



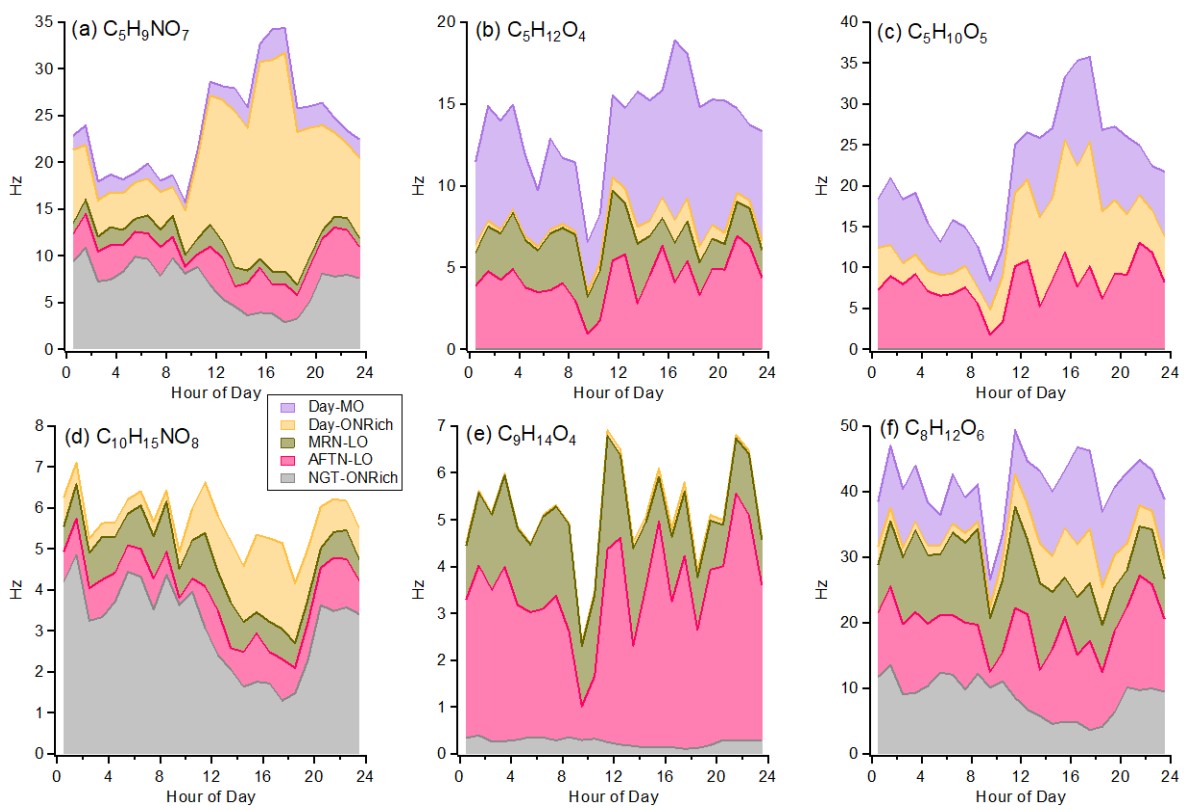

**Figure 6 Diurnal data of selected tracer species for isoprene and monoterpene SOA. (a) C₅H₉NO₇ (isoprene+NO₃·, isoprene+OH·+NOₓ); (b) C₅H₁₂O₄ (isoprene+OH·, IEPOX uptake); (c) C₅H₁₀O₅ (isoprene+OH·, non-IEPOX pathway); (d) C₁₀H₁₅NO₈ (α-/β-pinene+NO₃·, α-/β-pinene+OH·+NOₓ); (e) C₉H₁₄O₄ (fresh monoterpene SOA); (f) C₈H₁₂O₆ (aged monoterpene SOA).**





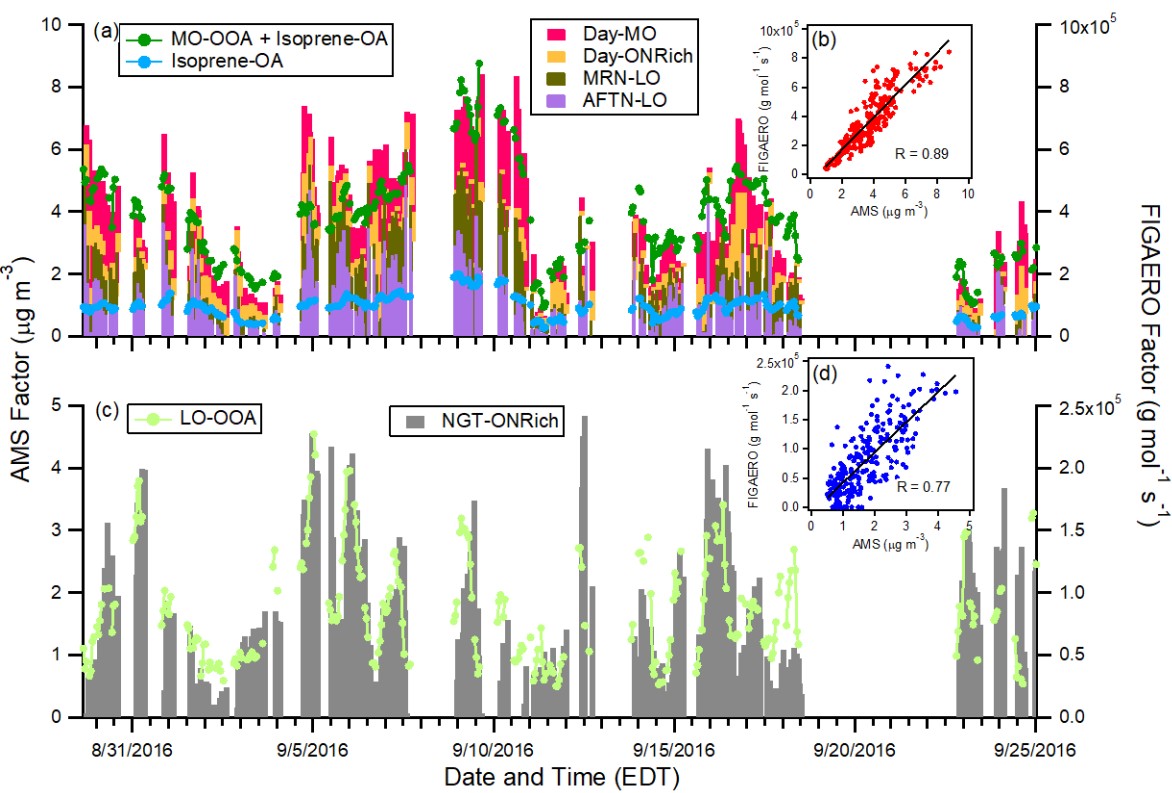

**Figure 7 Comparison between AMS daytime factors and FIGAERO-CIMS daytime factors ((a), (b)), and AMS nighttime factor and FIGAERO-CIMS nighttime factor ((c), (d)).**



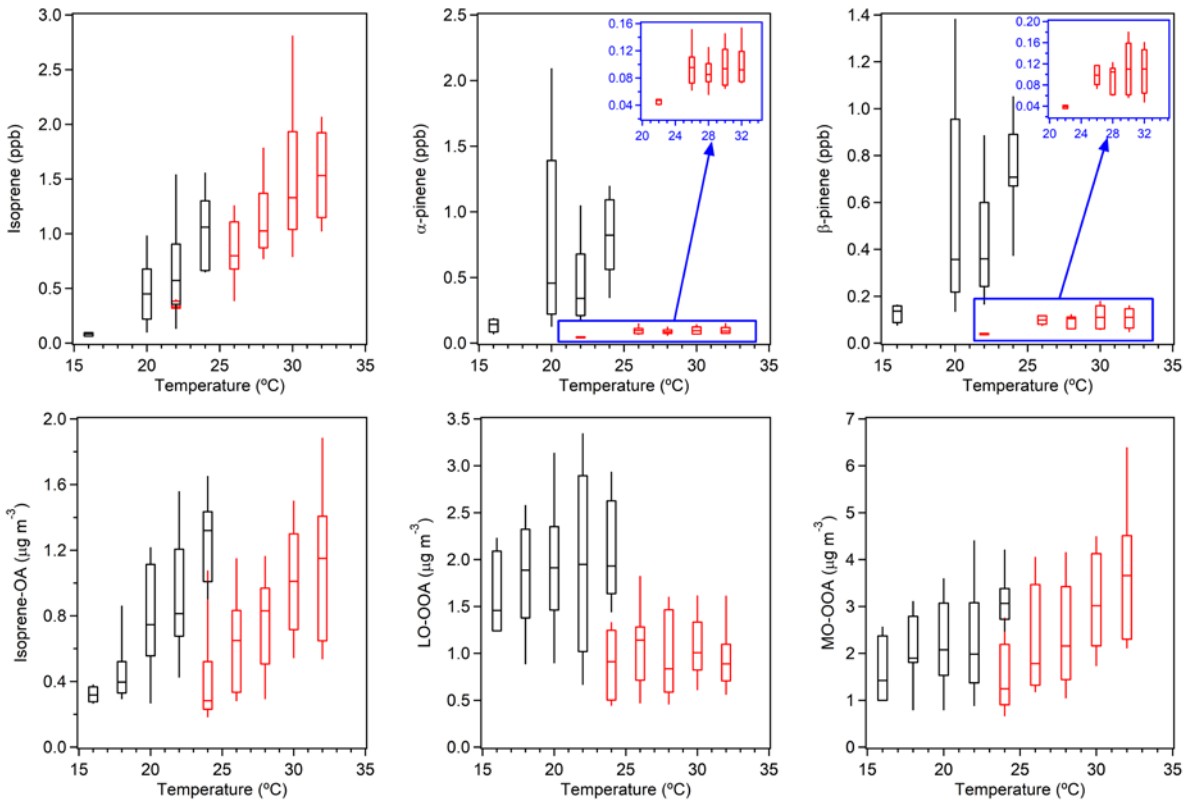

**Figure 8 Main biogenic VOC mixing ratios and AMS OA factor mass concentrations as a function of temperature. The data points are grouped into different temperature bins with a 2 °C increment and colored by time of day, where afternoon (12:00 – 16:00) measurements are in red and night (00:00 – 04:00) measurements are in black. The mid-point line, lower and upper boxes, lower and upper whiskers, represent median, 25th percentiles, 75th percentiles, 10th percentiles, and 90th percentiles, respectively.**





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
