# Peer review of "Chemical Characterization of Secondary Organic Aerosol at a Rural Site in the Southeastern U.S.: Insights from Simultaneous HR-ToF-AMS and FIGAERO-CIMS Measurements"

_Atmospheric Chemistry and Physics, 2020_

## Referee Comment (RC1) · Yunle Chen et al. · 7 Apr 2020

Chemical Characterization of SOA at a Rural site in the SE US: Insights from simultaneous HR-ToF-AMS and FIGAERO-CIMS measurements Chen et al.,

This paper describes the simultaneous use of FIGAERO-CIMS and AMS to carry out improved characterisation of the sources and transformation of organic aerosol at a rural site in the SE US. PMF has been used to split the mass spectra obtained by the two online methods and characterized factors that could be attributed to isoprene and

monoterpenes and with day-night separation. This is an interesting and well described study and shows the power of combining multiple instruments to obtain additional insights into SOA formation mechanisms. I recommend publication after the following minor points are addressed. I would also request some additional figures in the SI as the multi-part figures in the paper make it quite difficult to see the trends and detail (outlined below).

Minor comments Page 6, line 18: I think you need to make it clearer here that this unit is not an actual mass concentration, rather it is a scalar of the ion signal based on MW. Page 8, line 18: The NO levels are very low with a large error. Is the chemiluminescene detector use capable of giving accurate signals at these low mixing ratios? Page 9, line 15: The assumption here is that all the signal for CHON can be attributed to nitrates (-ONO2) rather nitrophenols (OH + NO2), which can also have three or more O and an odd number of H. I realise this is a rural site and therefore biogenic SOA may dominate but is there any evidence that nitrophenols are not important (i.e. biomass burning sources?). Also, is there evidence of how nitrophenols fragment in the AMS? Leading on from this, on page 10, line 27, you split into C6-C10. Are you sure there is no contribution to C6-C8 from nitrophenol type compounds? Page 15, line 23: Do you have an idea of the fraction of isoprene RO2 reacting with NO? Multiple times you discuss formation of products in the presence of NOx and it would be useful to estimate the fate of the isoprene peroxy radical in your field conditions. Page 17, lines 10-20: It is not very clear to my why you have chosen these particular isomers as being the "likely" source of these ions? The same is true on page 18, line 1-6. Surely there are multiple possible MT oxidation products at this formula? Are there any speciated MT VOC data to back this up? Im not sure why these ions are evidence of "aqueous processing"

Specific comments Page 7, line 15: can you suggest here why this data is different and needs the ME-2 approach to be used? Page 9, figure 1: What does the difference between the dotted and solid red lines tell you about the other N-containing species in

the aerosol? Also for N:C, I cant see what the trend in the dotted line is at this scale. In the O:C plot, the FIGAERO diurnal does show some enhancement in the afternoon but it is not entirely convincing. Are there a-typical days that cause this average to be more noisy? Page 10, line 18: I don't think table 1 has been mentioned before this point. I think it should be introduced earlier. Page 12, line 16: I feel this sentence just hangs there – can you clarify what you are using the nitrate tracer for LO-OOA to do? Page 13, line 2: I cant see the diurnal variability of the isoprene-OA in Figure 3, looks like a fairly straight line on this scale. Page 17: Is there any evidence of OS or NOS formation from the FIGAERO? Page 18, line 7: could it be that you are underestimating the amount of IEPOX-SOA using the FIGAERO because it is often converted to an OS? Page 19, line 1: The $C_5H_9NO_7$ peaks at 6pm. Is this still daylight? Page 19, line 10: Does this species correlate with NO? Page 19, line 26: Is it possible that $C_8H_{12}O_6$ is not a unique tracer for MT SOA? Page 21, line 1: Does a R=0.71 reall show they are "well correlated"? Page 22, line 7: The sentence starting "this can be explained" is hard to follow. Please rewrite.

Figure 2: What is going on in figure 2b in the morning? Seems very spiky. Figure 2e and 2f and in a few other places: I can see why you would put all the diurnals on one plot, but it is impossible to see if there is any trend in the minor factors. I would like to see additional figures in the SI with more appropriate axes. Figure 3 and 4: It is hard to see what is actual data in the time series. Remove the interpolation across missing data points. Figure 4: I would like to see a bigger version of the MS in the SI, with the major ions labelled. It is really hard to tell what ions are present. I would also suggest a table with the top 20 ions for each factor. Figure 6: It is really hard to see the true trends in each tracer when the factors are placed on top of each other. Can you add a figure to the SI showing the individual diurnals for each species? (i.e. similar to figure 4d) Page 15, line 2: I would like to see Figure 5 as a % of total signal plot in the SI.

---

## Referee Comment (RC2) · Anonymous Referee #2 · 9 Apr 2020

General comments

This paper describes the simultaneous use of an AMS and a FIGAERO-CIMS to measure SOA at a rural site in the southeastern USA. The chemical composition of the SOA detected is analysed via both techniques in terms of general chemical trends and where possible for specific chemical species that are known to be markers for SOA produced via different pathways. PMF based analysis of the AMS data and, most in-

terestingly, of the FIGAERO-CIMS data is used to identify and analyse factors that correspond to SOA produced from different starting VOCs and via different chemical pathways. Correlations between AMS and FIGAERO-CIMS factors are also explored.

This study addresses the topical scientific question of SOA formation mechanisms utilising the fairly novel application of PMF to FIGAERO-CIMS data in order to reach conclusions about the origin of SOA in the studied region in a well structured and clear manner. On this basis I recommend publication after a number of minor issues are addressed which I discuss below.

Specific comments

Page 4 Line 24: A little more detail on the influence on biogenic VOC emissions here would be welcome.

Page 6 Lines 10 - 11: What uniform sensitivity was used? Also how was the sensitivity arrived at? More details are needed here.

Page 6 Lines 16 - 19: I more detailed explanation of the validity and reasoning behind the method used here is needed as (in agreement with the other reviewers comments) I think this section is currently misleading/confusing.

Page 7 Line 13 and 18: The error on the alpha pinene measurement is so large as to make "negative" concentrations possible and this is even more apparent for the NO measurement. As the other reviewer has stated are such values ok?

Page 9 Line 7 - 10: When discussing the FIGAERO-CIMS data in Figure 1 I think the others are overstating when they say it is in "agreement" with the AMS data. Particularly in Figure 1A and 1C the random variation in the signal is larger than any perceived trend. Echoing comments from the other reviewer why is this so noisy?

Page 9 Line 25 - 27: In a similar vein when discussing Figure 2B I think the seemingly random variation in the signal at night makes interpretation problematic. Why is there so much variation? Is it because the overall signal is very low? Would like to see the

"actual" diurnals with the real signal rather than just the proportion diurnals to assess this further.

Page 16 Line 21 - 24: I would like to know more about how the quoted TMAX values were obtained? Looking at Figure S10 (and particularly Figure S10B) it is hard to see how those precise value were chosen, with a seemingly more obvious peak at lower T not identified?

Technical comments

The acronym SOA should be defined where it is first used (like all the others are) rather than later on as it is currently.

Page 10 Line 3: Convert not covert.

Page 20 Line 12: Prevalent not prevalence.

---

## Author Comment (AC1) · 2 Jun 2020

The comment was uploaded in the form of a supplement:
https://www.atmos-chem-phys-discuss.net/acp-2020-126/acp-2020-126-AC1-supplement.pdf

2020.

---

## Author Response (AR1)

We thank the reviewers for the insightful comments. We have addressed the reviewers' comments point by point as indicated below and revised the manuscript accordingly. The reviewers' comments are in italics and changes made to the manuscript are in quotation marks.

**Reviewer #1**

**General comments:**

*This paper describes the simultaneous use of FIGAERO-CIMS and AMS to carry out improved characterisation of the sources and transformation of organic aerosol at a rural site in the SE US. PMF has been used to split the mass spectra obtained by the two online methods and characterized factors that could be attributed to isoprene and monoterpenes and with day-night separation. This is an interesting and well described study and shows the power of combining multiple instruments to obtain additional insights into SOA formation mechanisms. I recommend publication after the following minor points are addressed. I would also request some additional figures in the SI as the multi-part figures in the paper make it quite difficult to see the trends and detail (outlined below).*

**Response:** We thank the reviewer for the thoughtful comments. We added the additional figures to SI and revised the manuscript to address the specific comments, as indicated below.

**Specific comments:**

1) *Page 6, line 18: I think you need to make it clearer here that this unit is not an actual mass concentration, rather it is a scalar of the ion signal based on MW.*
   **Response:** We have clarified this in the revised manuscript (Page 6 Line 24 of the revised manuscript): "It is noted that the unit g mol$^{-1}$ s$^{-1}$ is a scalar of the ion signal based on MW and not an actual mass concentration."

2) *Page 8, line 18: The NO levels are very low with a large error. Is the chemiluminescene detector use capable of giving accurate signals at these low mixing ratios?*

   **Response:** The large standard error of the mean reflects the strong diurnal variations (rather than measurement errors) as shown in Figure R1 below. The chemiluminescene detector was capable of giving accurate signals at low NO levels. NO level was generally low with strong diurnals at Yorkville during our measurements. We clarified in the revised manuscript when we first used average ± SE to describe the data (Page 8 Line 15 of the revised manuscript):

…the two-month measurements were characterized by moderate temperature (24.0 ± 4.0 °C, average ± SE if not specified hereafter)….

[Figure]

**Figure R1** Diurnal profile of NO.

3) *Page 9, line 15: The assumption here is that all the signal for CHON can be attributed to nitrates (-ONO2) rather nitrophenols (OH + NO2), which can also have three or more O and an odd number of H. I realise this is a rural site and therefore biogenic SOA may dominate but is there any evidence that nitrophenols are not important (i.e. biomass burning sources?). Also, is there evidence of how nitrophenols fragment in the AMS? Leading on from this, on page 10, line 27, you split into C6-C10. Are you sure there is no contribution to C6-C8 from nitrophenol type compounds?*

**Response:** AMS factorization analysis did not resolve any biomass burning OA factor and there is no known local source of biomass burning, although we cannot rule out the presence of aged biomass burning at Yorkville transported from elsewhere (e.g., as a component of MO-OOA). Aromatic compounds were detected but were much less abundant than biogenic VOC at Yorkville (Figure R2). Meanwhile, iodide reagent ion tends to have a higher sensitivity towards polar and oxygenated organic compounds (Lee et al., 2014), and thus a higher sensitivity towards organic nitrates compared to nitrophenols was expected for FIGAERO-CIMS measurements. Taken together, while we cannot completely rule out the presence of nitrophenols at Yorkville, they are not the major contributors to the measured signals and should not change the conclusions of current work.

[Figure]

**Figure R2** Campaign-average mixing ratios of measured VOC.

4) *Page 15, line 23: Do you have an idea of the fraction of isoprene RO2 reacting with NO? Multiple times you discuss formation of products in the presence of NOx and it would be useful to estimate the fate of the isoprene peroxy radical in your field conditions.*

**Response:** We did not include branching ratios in the discussion as radical measurements (e.g., $HO_2$) were not available at Yorkville. However, given the level of NO measured at the site, the reaction of isoprene $RO_2$ with NO was expected. At Yorkville, NO was the highest at 9-10 a.m. with a mixing ratio of ~ 600 ppt, and below 100 ppt for most of the day (**Figure R1**). Chen et al. (2017) measured $HO_2$ concentration at urban Atlanta in 2015 summer, which was the highest at ~ 2 p.m. with a mixing ratio of ~20 ppt, and low in the morning with a mixing ratio of near zero. If we just consider $RO_2$+NO vs $RO_2$+$HO_2$, using $k_{ISOPOO+NO}$ = 8.74e-12 $cm^3$ $molec^{-1}$ $s^{-1}$ and $k_{ISOPOO+HO2}$ = 1.74e-11 $cm^3$ $molec^{-1}$ $s^{-1}$ (Wennberg et al., 2018), $RO_2$ exclusively reacted with NO at 9 a.m.when NO peaked and $HO_2$ was low, and the branching between NO and $HO_2$ fates was ~ 2:1 at 2 p.m. when $HO_2$ peaked and NO was low. Although Yorkville is cleaner than urban Atlanta and the $HO_2$ level could be higher, the NO level at the site dictated that $RO_2$+NO reaction proceeded in the presence of $NO_x$.

5) *Page 17, lines 10-20: It is not very clear to my why you have chosen these particular isomers as being the "likely" source of these ions? The same is true on page 18, line 1-6. Surely there are multiple possible MT oxidation products at this formula? Are there any speciated MT VOC data to back this up? Im not sure why these ions are evidence of "aqueous processing".*

**Response**: We agree that with only chemical formula, it is hard to unambiguously determine the structures or sources of these ions. Hence, we discussed possible identities of these ions based on previous publications and our understanding of the chemistry (biogenic VOC) at the site. For example,

as we discussed in the original manuscript, for the ion $C_4H_4O_6$, it has been reported as 2-hydroxy-3-oxosuccinic acid, a product of OH· initiated oxidation of aqueous succinic and tartaric acids (Chan et al., 2014;Cheng et al., 2016). In addition, the formula implies that it likely has more than one carboxylic acid group, and generally di- and tricarboxylic acids are more likely to form in aqueous processing. Therefore, we tentatively assumed that $C_4H_4O_6$ was formed in aqueous processing.

6) *Page 7, line 15: can you suggest here why this data is different and needs the ME-2 approach to be used?*

**Response:** We discussed possible causes in detail in Section 3.3 of the original manuscript. As discussed, the contribution from Isoprene-OA was largely overestimated in the unconstrained PMF, likely due to interferences as measurements were conducted during transition in seasons (and isoprene emissions). We clarified this by modifying the sentence "A description of our ME-2 analysis is provided in Section 3" to "A description of our unconstrained PMF and ME-2 analyses is provided in Section 3.3" (Page 7 Line 25 of the revised manuscript).

7) *Page 9, figure 1: What does the difference between the dotted and solid red lines tell you about the other N-containing species in the aerosol? Also for N:C, I cant see what the trend in the dotted line is at this scale.*

**Response:** While the nitrate functionality of organic nitrates largely fragments into $NO^+$ and $NO_2^+$ in the AMS, some of organic nitrates can still be measured as CHON ions (Farmer et al., 2010). Reduced nitrogen species (e.g., amines) also produce CHN and CHON signals in AMS (Ge et al., 2014). These could be two contributors to dotted N:C ratio. The N:C ratio without accounting for the $NO^+$ and $NO_2^+$ fragments produced by organic nitrates was too low and failed to show much trend so we did not discuss more about it.

8) *In the O:C plot, the FIGAERO diurnal does show some enhancement in the afternoon but it is not entirely convincing. Are there a-typical days that cause this average to be more noisy?*

**Response:** The more noisy FIGAERO elemental ratio diurnals did not seem to be caused by one or two a-typical days. One possible explanation is that the uniform sensitivity assumption may allow some less important ions, whose actual concentrations were lower than that have been accounted for in the analysis, to have a larger impact on determining the average elemental ratios, and vice versa.

9) *Page 10, line 18: I don't think table 1 has been mentioned before this point. I think it should be introduced earlier.*

**Response:** We thank the reviewer for catching this. We have added introduction of Table 1 at the beginning of the paragraph (Page 10 Line 5 of the revised manuscript):

"On average, pOC and pON contributed 87.7 ± 10.8 % and 12.3 ± 10.8 %, respectively, to total FIGAERO-CIMS signals (Table 1)".

10) *Page 12, line 16: I feel this sentence just hangs there – can you clarify what you are using the nitrate tracer for LO-OOA to do?*

**Response:** In PMF analysis, different external tracers were used for identifying OA factors. Here, we discussed external tracers for Isoprene-OA and LO-OOA. In Page 12 Line 6 of the original manuscript we mentioned that "the determination of a final solution was guided by three criteria: mass fraction of each factor (Figure S5(b)), correlation between factor time series with external tracers, and the $f_{C5H6O}$ of resolved Isoprene-OA (Figure S5(c))". In the following sentences we introduced the external tracer for Isoprene-OA (C5H12O4, Page 12 Line 8-14 of the original manuscript), and the external tracer for LO-OOA (organic nitrate functionality, Page 12 Line 14-16 of the original manuscript). Previous studies (Xu et al., 2015) showed that organic nitrates made up a large portion of LO-OOA and correlated well with LO-OOA. Hence, we used organic nitrate functionality as an external tracer for LO-OOA.

11) *Page 13, line 2: I cant see the diurnal variability of the isoprene-OA in Figure 3, looks like a fairly straight line on this scale.*

**Response:** We thank the reviewer for catching this. We have modified Figure 3d so that the trends of minor factors can be seen clearly.

12) *Page 17: Is there any evidence of OS or NOS formation from the FIGAERO?*

**Response:** We did not see much OS or NOS signals. In addition, the low-volatility OS and NOS products could decompose during thermal desorption processes and lose their functionalities (Lopez-Hilfiker et al., 2016).

13) *Page 18, line 7: could it be that you are underestimating the amount of IEPOX-SOA using the FIGAERO because it is often converted to an OS?*

**Response:** We did not specifically estimate the amount of IEPOX-SOA through FIGAERO analysis. As mentioned in our response to comment #12, the OS products could decompose during thermal desorption processes (Lopez-Hilfiker et al., 2016).

14) *Page 19, line 1: The C5H9NO7 peaks at 6pm. Is this still daylight?*

**Response:** Yes, sunset was after ~7pm during our measurements. The $C_5H_9NO_7$ peak at 6pm was more like an accumulation of daytime formation.

15) *Page 19, line 10: Does this species correlate with NO?*
**Response:** No, this species did not directly correlate with NO (R = -0.12).

16) *Page 19, line 26: Is it possible that C8H12O6 is not a unique tracer for MT SOA?*
**Response:** It is possible, but to the best of our knowledge, $C_8H_{12}O_6$ is a well-established tracer for aged MT SOA (Szmigielski et al., 2007;Zhang et al., 2010;Müller et al., 2012;Eddingsaas et al., 2012), and has not been reported to form in significant abundance from other precursors. Given biogenic VOC dominated at Yorkville, it was likely from MT SOA.

17) *Page 21, line 1: Does a R=0.71 reall show they are "well correlated"?*
**Response:** We have changed from "well correlated" to "mildly correlated" in the revised manuscript (Page 21 Line 9 of the revised manuscript).

18) *Page 22, line 7: The sentence starting "this can be explained" is hard to follow. Please rewrite.*
**Response:** We have rewritten the sentence in the revised manuscript (Page 22 Line 16 of the revised manuscript):
"This can be explained by the longer lifetime of aerosol compared to gas species, and as a result the high concentration of nighttime Isoprene-OA was residue from daytime formation."

19) *Figure 2: What is going on in figure 2b in the morning? Seems very spiky.*
**Response:** It was possibly caused by pON spike at ~ 10am. Even Yorkville is a rural site, we observed a morning NO peak and hypothesized that it resulted from NO transported from nearby roads (Figure R1and Page 8 Line 18 of the original manuscript). We have clarified it in the revised manuscript (Page 10 Line 8 of the revised manuscript):
"A 10 am peak was also observed for pON fraction, following a NO peak at around 9 am, likely due to enhanced ON formation as NO level increased."

20) *Figure 2e and 2f and in a few other places: I can see why you would put all the diurnals on one plot, but it is impossible to see if there is any trend in the minor factors. I would like to see additional figures in the SI with more appropriate axes.*
**Response:** We have modified Figure 2e and 2f so that the trends of minor factors can be seen clearly.

21) *Figure 3 and 4: It is hard to see what is actual data in the time series. Remove the interpolation across missing data points.*

   **Response:** We have modified Figures 3 and 4 to show actual time series better.

22) *Figure 4: I would like to see a bigger version of the MS in the SI, with the major ions labelled. It is really hard to tell what ions are present. I would also suggest a table with the top 20 ions for each factor.*

   **Response:** We have added Figure S10 and Table S1 to the revised SI.

23) *Figure 6: It is really hard to see the true trends in each tracer when the factors are placed on top of each other. Can you add a figure to the SI showing the individual diurnals for each species? (i.e. similar to figure 4d)*

   **Response:** The factors were not placed on top of each other, but added on top of each other. Therefore the diurnal of each tracer (total signal) was shown as the top line in each figure. If you were referring to diurnals of each species in each factor, the trend would agree with diurnal of the factor itself (Figure 4d), given how we calculated it.

24) *Page 15, line 2: I would like to see Figure 5 as a % of total signal plot in the SI.*

   **Response:** We added Figure S11 to the revised SI.

**Reviewer #2**

**General comments:**

*This paper describes the simultaneous use of an AMS and a FIGAERO-CIMS to measure SOA at a rural site in the southeastern USA. The chemical composition of the SOA detected is analysed via both techniques in terms of general chemical trends and where possible for specific chemical species that are known to be markers for SOA produced via different pathways. PMF based analysis of the AMS data and, most interestingly, of the FIGAERO-CIMS data is used to identify and analyse factors that correspond to SOA produced from different starting VOCs and via different chemical pathways. Correlations between AMS and FIGAERO-CIMS factors are also explored. This study addresses the topical scientific question of SOA formation mechanisms utilizing the fairly novel application of PMF to FIGAERO-CIMS data in order to reach conclusions about the origin of SOA in the studied region in a well structured and clear manner. On this basis I recommend publication after a number of minor issues are addressed which I discuss below.*

**Response:** We thank the reviewer for the thoughtful comments. We revised the manuscript to address the specific comments as indicated below.

**Specific comments:**

1) *Page 4 Line 24: A little more detail on the influence on biogenic VOC emissions here would be welcome.*
   **Response:** In the original manuscript, we discussed the VOC emissions in later sections but we agree it would be helpful to add some details here too. We have clarified it in the revised manuscript (Page 4 Line 25 of the revised manuscript):
   "…which had a direct influence on biogenic VOC emissions, e.g., isoprene mixing ratio decreased from more than 2 ppb at the beginning of the campaign to below 1 ppb at the end (daily average)."

2) *Page 6 Lines 10 - 11: What uniform sensitivity was used? Also how was the sensitivity arrived at? More details are needed here.*
   **Response:** No sensitivity conversion (normally in the unit of Hz ppt$^{-1}$) was applied to the FIGAERO-CIMS dataset. The signals are in the unit of Hz, and therefore all ions had a same sensitivity. We rewrote the sentence to clarify this point in the revised manuscript (Page 6 Line 12 of revised manuscript):
   "The signals reported are in counts per second (Hz), if not specified in the following discussion. As no further sensitivity conversion is applied to the data, reporting the data in Hz implicitly assumes a uniform sensitivity without considering the magnitude of sensitivity conversion factor for FIGAERO-CIMS measurements."

3) *Page 6 Lines 16 - 19: I more detailed explanation of the validity and reasoning behind the method used here is needed as (in agreement with the other reviewers comments) I think this section is currently misleading/confusing.*
   **Response:** As noted in our response to Reviewer #1 comment #1, we have clarified in the revised manuscript (Page 6 Line 24 of revised manuscript):
   "It is noted that the unit g mol$^{-1}$ s$^{-1}$ is a scalar of the ion signal based on MW and not an actual mass concentration."

4) *Page 7 Line 13 and 18: The error on the alpha pinene measurement is so large as to make "negative" concentrations possible and this is even more apparent for the NO measurement. As the other reviewer has stated are such values ok?*

**Response:** The large standard error of the mean reflects the strong diurnal variations (rather than measurement errors). Please refer to response to Reviewer #1 comment #2 and Figure R1.

5) *Page 9 Line 7 - 10: When discussing the FIGAERO-CIMS data in Figure 1 I think the others are overstating when they say it is in "agreement" with the AMS data. Particularly in Figure 1A and 1C the random variation in the signal is larger than any perceived trend. Echoing comments from the other reviewer why is this so noisy?*

   **Response:** We thank the reviewer for the comments. We did not use "agreement" or mean to imply this when describing Figure 1 in the original manuscript. We recognized and discussed the discrepancies between AMS and FIGAERO-CIMS elemental ratio diurnals. As our response to Reviewer #1 comment #8, the more noisy FIGAERO elemental ratio diurnals are possibly due to that the uniform sensitivity assumption may allow some less important ions, whose actual concentrations were lower than that have been accounted for in the analysis, to have a larger impact on determining the average elemental ratios.

6) *Page 9 Line 25 - 27: In a similar vein when discussing Figure 2B I think the seemingly random variation in the signal at night makes interpretation problematic. Why is there so much variation? Is it because the overall signal is very low? Would like to see the "actual" diurnals with the real signal rather than just the proportion diurnals to assess this further.*

   **Response:** As our response to reviewer #1 comment #19, the 10 am spike was possibly caused by pON formation following the NO morning peak. The overall signals of either pOC or pON were not too low. pOC and pON contributed 87.7 ± 10.8 % and 12.3 ± 10.8 % on average (Page 9 Line 24 of the original manuscript). One possible explanation is that the uniform sensitivity assumption may allow some less important ions, whose actual concentrations were lower than that have been accounted in the analysis, to have a larger contribution to overall signals. The actual diurnals of pOC and pON are shown in Figure R3.

[Figure]

**Figure R3** Diurnal profiles of total pOC and total pON signals.

7) *Page 16 Line 21 - 24: I would like to know more about how the quoted TMAX values were obtained? Looking at Figure S10 (and particularly Figure S10B) it is hard to see how those precise value were chosen, with a seemingly more obvious peak at lower T not identified?*

**Response:** We thank the reviewer for this careful insight. We realized that Figure S10 was not correctly made in the original manuscript: 1) Figure S10b should be $C_3H_4O_4$ and Figure S10C should be $C_3H_4O_5$, 2) $C_3H_4O_4$ and $C_3H_4O_5$ had three resolved $T_{max}$ rather than two, which have been corrected in the revised manuscript. We also added fitted peaks to revised Figure S10. We did the $T_{max}$ fitting following methods described in Stark et al. (2017). Specifically, when fitting the thermogram, we tried to minimize the number of peaks required to explain the curve while certain residues were allowed, so the fitted curve did not perfectly agree with the measured one.

8) *The acronym SOA should be defined where it is first used (like all the others are) rather than later on as it is currently.*

**Response:** We have defined SOA in the introduction when it is first mentioned (Page 2 Line 14 of the revised manuscript):

"…are the most dominant biogenic VOC and secondary OA (SOA) precursors…"

9) *Page 10 Line 3: Convert not covert.*

**Response:** We have changed covert to convert.

10) *Page 20 Line 12: Prevalent not prevalence.*

**Response:** We have changed prevalence to prevalent.

[revised manuscript text omitted]

**Figure S10 (a) normalized mass spectra of each FIGAERO OA factor and (b) difference between normalized mass spectra of each FIGAERO OA factor and normalized overall average mass spectra.**

[Figure]

**Figure S11 Fraction of pOC and pON ions of different carbon numbers (grouped as C1-5, C6-10, C11-15, and C>15) in total FIGAERO-CIMS signals.**

[Figure]

**Figure S12 Thermograms of (a) C₅H₉NO₇, (b) C₃H₄O₅, and (c) C₃H₄O₄ ions measured by FIGAERO-CIMS.**

**Table S1 Most abundant 20 species of each FIGAERO factor listed in order of abundance.**

|  | Day-MO | Day-ONRich | MRN-LO | AFTN-LO | NGT-LO |
|---|---|---|---|---|---|
| 1 | C4H4O6 | C4H6O5 | C8H12O5 | C4H4O6 | C4H6O5 |
| 2 | C4H6O5 | C3H4O5 | C3H4O4 | C5H8O5 | C8H12O5 |
| 3 | C5H6O6 | C5H8O5 | C4H6O5 | C8H12O5 | C7H10O6 |
| 4 | C5H8O6 | C3H4O4 | C5H8O5 | C4H6O5 | C4H4O6 |
| 5 | C5H8O5 | C5H9NO7 | C7H10O5 | C5H6O5 | C8H10O6 |
| 6 | C8H12O5 | C5H6O5 | C7H8O5 | C5H8O6 | C3H4O5 |
| 7 | C5H6O5 | C6H8O5 | C2H2O4 | C7H10O5 | C8H12O6 |
| 8 | C6H8O6 | C6H8O6 | C8H10O5 | C7H10O6 | C5H6O6 |
| 9 | C3H4O5 | C8H12O5 | C8H10O6 | C9H14O5 | C8H10O5 |
| 10 | C7H10O6 | C7H10O6 | C5H6O5 | C5H6O5 | C7H8O5 |
| 11 | C8H10O6 | C5H10O5 | C7H10O6 | C8H12O6 | C3H4O4 |
| 12 | C5H4O5 | C6H10O5 | C6H8O5 | C8H10O6 | C5H9NO7 |
| 13 | C7H8O6 | C7H8O5 | C6H8O6 | C6H10O5 | C2H2O4 |
| 14 | C6H8O5 | C5H4O5 | C5H6O6 | C6H8O5 | C7H10O5 |
| 15 | C2H4O3 | C3H2O4 | C9H14O5 | C8H10O5 | C5H8O6 |
| 16 | C6H6O6 | C7H10O5 | C8H12O6 | C6H8O6 | C9H12O6 |
| 17 | C6H8O7 | C1H2O2 | C6H10O5 | C7H12O5 | C5H6O5 |
| 18 | C7H8O5 | C5H6O6 | C9H12O6 | C5H10O5 | C10H14O6 |
| 19 | C2H2O4 | C5H8O6 | C7H8O6 | C9H12O6 | C7H8O6 |
| 20 | C7H10O5 | C5H7NO7 | C6H6O5 | C2H4O3 | C5H8O5 |